# A model of early-life interactions between the gut microbiome and adaptive immunity provides insights into the ontogeny of immune tolerance

Burcu Tepekule[1]*, Ai Ing Lim[2], Charlotte Jessica E. Metcalf[1]*

1 Department of Ecology and Evolutionary Biology, Princeton University, Princeton, New Jersey, United States of America, 2 Department of Molecular Biology, Princeton University, Princeton, New Jersey, United States of America

\* burcutepekule@gmail.com (BT); cjmetcalf@princeton.edu (CJEM)

## Abstract

To achieve immune and microbial homeostasis during adulthood, the developing immune system must learn to identify which microbes to tolerate and which to defend against. How such 'immune education' unfolds remains a major knowledge gap. We address this gap by synthesizing existing literature to develop a mechanistic mathematical model representing the interplay between gut ecology and adaptive immunity in humans during early life. Our results indicate that the inflammatory tone of the microenvironment is the mediator of information flow from pre- to post-weaning periods. We evaluate the power of postnatal fecal samples for predicting immunological trajectories and explore breastfeeding scenarios when maternal immunological conditions affect breastmilk composition. Our work establishes a quantitative basis for 'immune education', yielding insights into questions of applied relevance.

## Introduction

The global burden of immune-mediated disease is rapidly growing [1]. Epidemiological data indicate that early life exposures are key determinants of immune-mediated diseases later in life [2], such as allergies [3], asthma [4], type 1 diabetes [5], and inflammatory bowel disease (IBD) [6]. This impact of early life is primarily attributed to interactions between the microbiome and the immune system during a critical developmental window, which enables hosts to establish tolerance to commensal bacteria [7], ensuring the maintenance of immune and microbial homeostasis into adulthood, while appropriately defending against pathogens [8,9]. When this crosstalk is perturbed, pathological imprinting may develop, characterized by excessive immune reactivity and increased susceptibility to inflammatory diseases in adulthood [10]. Both inherent microbial factors and maternal cues determine how microbe-immune interactions unfold during early life: symbiotic commensals provide metabolic products that establish regulatory pathways facilitating a balanced immune response,

**Data availability statement:** All data used in this study are publicly available through the sources cited in the manuscript. The corresponding code and curated datasets are and code used in this work is available at https://doi.org/10.5281/zenodo.15629746 under the GNU General Public License v3.0.

**Funding:** This work was supported by Schweizerischer Nationalfonds zur Förderung der Wissenschaftlichen Forschung (SNSF) postdoc mobility grant P500PB_206889 (to BT); by Branco Weiss Fellowship—Society in Science administered by the ETH Zurich and Rutgers Cancer Institute of New Jersey New Investigator Award (to AIL); Princeton Catalysis Institute Grant (to CJEM) and Princeton Precision Health Grant (to CJEM). The funders had no role in study design, data collection and analysis, decision to publish, or preparation of the manuscript.

**Competing interests:** The authors have declared that no competing interests exist.

**Abbreviations:** ABL, Assumed Based on Literature; AUC, area under the receiver operating characteristic curve; BCR, B-cell receptor; DCs, dendritic cells; ECF, Exclusive Complementary Feeding; eSIgA, endogenous SIgA; GALT, gut-associated lymphoid tissues; GC, germinal center; gLV, generalized Lotka–Volterra; HMOs, human milk oligosaccharide; IBD, inflammatory bowel disease; LP, Lamina Propria; M, Microfold; mSIgA, maternal secretory Immunoglobulin A; PDPs, plant-derived polysaccharides; PRRs, pattern recognition receptors; P-SHM-S, proliferation, somatic hypermutation, and selection; SIgA, secretory Immunoglobulin A; Tfh, T follicular helper; Tfr, T follicular regulatory.

whereas pathogens stimulate the immune system to develop defense mechanisms. Breastmilk delivers microbes and nutrients, but also antibodies that dictate the timing and nature of bacterial antigen presentation to the infant's developing immune system, establishing a transgenerational cycle of immune priming [11,12]. The collective influence of these processes on immune ontogeny and maturation is encapsulated by the term 'immune education' [13].

The information available to tackle the establishment of 'immune education' is now considerable [14]. To date, thousands of papers have been published, ranging from observational data from human populations to experimental perturbations in animal models. However, experimental methods inevitably focus on a limited set of mechanisms, while larger-scale descriptive analysis rooted in observational data may illuminate patterns, but ultimately yield verbal descriptions lacking mathematical characterization. The time is ripe to integrate these layers of evidence into a systems biology framework that formally accounts for the multiple potential interacting components. Such a foundation will open the way to generating testable hypotheses for empirical investigation based on experimental and clinical data, and further mechanistic modeling.

To this end, we introduce a mathematical framework describing the reciprocal imprinting of the human gut microbiome and the antigen-specific endogenous mucosal secretory Immunoglobulin A (SIgA) response during the first two years of life. While immune outcomes are undoubtedly multifaceted and influenced by numerous factors, many lines of evidence indicate the importance of the SIgA response, as its reactivity to microbial antigens plays a crucial role in mediating gut immune responses and their dysregulation [15,16]. The SIgA response also integrates both the B cell and T-cell arms of immune ontogeny, and manifests the dual functionality of the gut mucosal immune system by selectively neutralizing pathogens while tolerating commensal bacteria beneficial for immune homeostasis [17,18]. Furthermore, SIgA represents the transgenerational aspect of immune priming: maternal SIgA modulates antigen presentation while simultaneously regulating the gut community composition [11,19].

Our mechanistic modeling strategy (Fig 1) blends flexibility with tractability in reflecting the effects of maternal factors, feeding practices, consumer resource dynamics within microbial communities, and multifunctionality of SIgA (Fig 1B); and introduces a quasi-stochastic mathematical model of germinal center (GC) reactions embedded in a combination of ordinary and delay differential equations (Fig 1C). We parameterize our framework using data from infant fecal samples [20–23], formally mapping microbial abundance measurements into expectations for the dynamics of the lumen and the GCs of the gut, contexts traditionally obscured by the impracticality of direct sampling [24] (Fig 1A). The mechanistic model structure entails a high-dimensional parameter space, but qualitative results are consistent across parameter combinations (S1 Text, sections Global and Local Sensitivity Analyses).

Overall, our model brings mathematical formalism to the concept of 'immune education'. This framework enables us to explore three questions with translational implications. First, it supports the quantification of potentially key diagnostic markers;

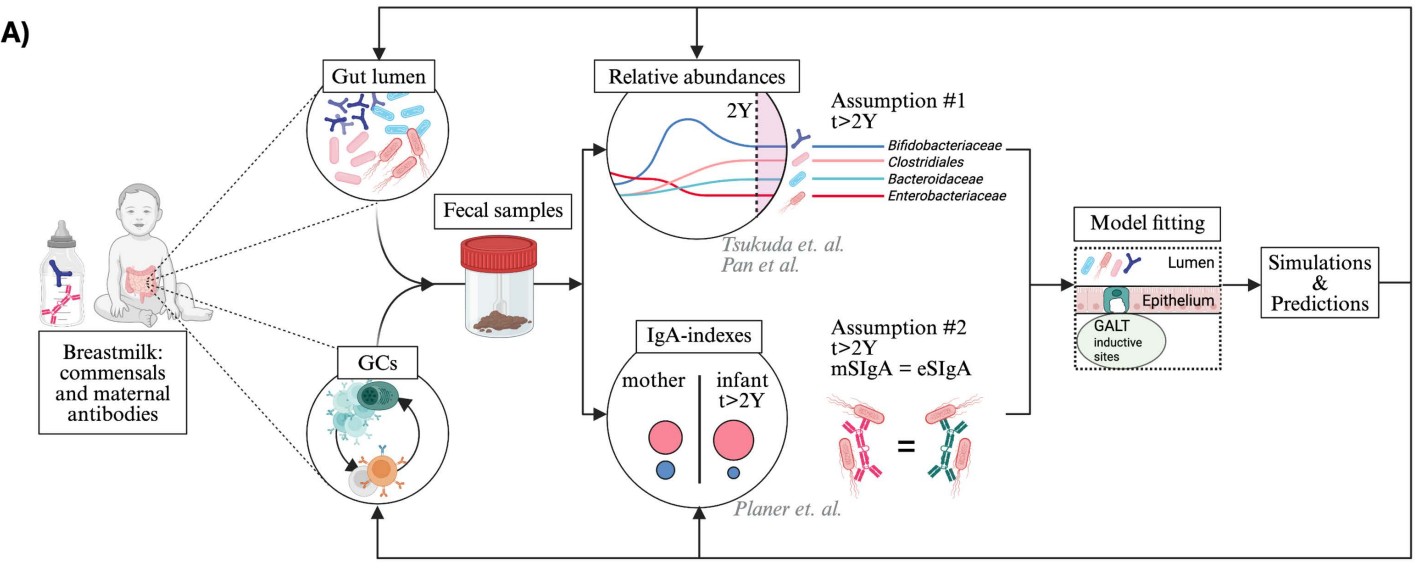

**Fig 1. Modeling framework. A)** Key data sources and assumptions used during model fitting, including relative abundance (line plot) and IgA-index (bubble plot) data, each colored by taxon. Model predictions reconstruct ecological dynamics in the gut lumen and the germinal center (GC) reactions.

2Y: 2 years; mSIgA and eSIgA: maternal and endogenous secretory Immunoglobulin A. **B)** Model dynamics. **(i)** Breastfeeding and complementary feeding introduce human milk oligosaccharides (HMOs), plant-derived polysaccharides (PDPs), **(ii)** mSIgA, and **(iii)** commensal bacteria to the gut lumen as SIgA-bacteria complexes or bacteria alone, alongside with the bacteria seeding the infant's gut during vaginal birth. The microbiome within the gut lumen (large light blue circle), contains subpopulations of commensals that exist either in an SIgA-bound (white circle, left) or unbound (white circle, right) state, and transition between the two is driven by SIgA coating or degradation. **(iv)** Bacterial populations modulate the inflammatory tone of the microenvironment of the gut lumen ($\sigma$), **(v)** influencing epithelial cell-mediated communication with the organized gut-associated lymphoid tissue (GALT) inductive sites, and **(vi)** the antigenic sampling through barrier integrity. **(vii)** SIgA-bound and unbound antigens sampled by Microfold (M) cells **(viii)** activate the naïve B cells, and **(ix)** drive the balance between antigen-specific regulatory vs. helper T cells (indicated by the divide sign). **(x)** This balance and $\sigma$ shape the selection threshold $\delta_i$ for taxon $i$, **(xi)** which determines the BCR affinity ranges that lead to different B cell fates: **(xii)** apoptosis, **(xiii)** continued circulation, or **(xiv)** terminal differentiation into IgA-secreting plasma cells. **(xv)** IgA secreted by the plasma cells regulates the microbiome via **(xvi)** masking and **(xvii)** neutralization functions. Alongside the HMOs and PDPs, **(xviii)** the oxygen concentration ($O_2$) also influences the ecological dynamics based on commensals' metabolism. **C)** GC reactions (steps 1–7, materials and methods, corresponding to **(x)**–**(xiv)** on the previous panel). Naïve B cells with a wide BCR affinity distribution migrate from the bone marrow to the GALT inductive sites, and their affinity distribution is combined with that of circulating B-cells. B-cells activate upon encountering bacterial antigens. Based on how close their BCR affinities are to the selection threshold $\delta_i$, cells undergo positive selection, receiving adequate T cell help to continue circulating or differentiate to IgA-secreting plasma cells; and if not, become apoptotic. Circulating cells undergo somatic hypermutation (SHM), increasing the standard deviation of their BCR affinity distribution, continuing the cycle.

second, the simulation of clinical intervention strategies, and third, identification of preventive measures before pathological trajectories are imprinted. Overall, it allows us to systematically explore the emergent properties of this complex system in a controlled, interpretable, and reproducible setting. In doing so, it opens the way to investigating persistent questions in early-life immunology with both fundamental and applied relevance.

## Results

### Model outcomes unveil the dynamics in gut lumen, successfully predicting out-of-sample data

Capturing immune education requires encompassing exogenous inputs into the gut lumen alongside endogenous dynamics in the gut lumen and organized gut-associated lymphoid tissue (GALT) inductive sites such as Peyer's patches and colonic patches (Fig 1A, see S4 Table for anatomical distinctions relevant to the model). Exogenous inputs quantified based on different feeding practices of infants [25–27] (Eq. 18–21), include caloric intake from human milk oligosaccharides (HMOs) and plant-derived polysaccharides (PDPs), and maternal secretory Immunoglobulin A (mSIgA) concentrations. HMOs and PDPs differentially modulate growth rates of different bacterial taxa based on their metabolism [28–32] (Eq. 4). Timing of endogenous immune system activation is dictated by the maturation of Microfold (M) cells, which serve as the primary route for antigen transport to GALT inductive sites (Fig 1B), the key site for the development of the gut adaptive immune response [33,34]. M cells are shown to appear shortly before weaning in mice models, primarily due to decreasing concentrations of maternal steroids in breastmilk [35], for which we use mSIgA as a proxy, given a lack of quantitative data. While the timing of M cell maturation in humans remains unclear, we address this uncertainty through sensitivity analysis (S1 Text, section Local Sensitivity Analyses). M cell maturation translates into antigenic sampling — and thus the start of immune response maturation (Fig 2A).

To fit our model to fecal microbiome data [20–23] and the established timeline of immune ontogeny [14,35], we leverage two observations: (*i*) the infant develops an SIgA profile resembling their nursing mother's [21], and (*ii*) the infant's immune response and microbial community structure stabilize after two years [33] (Fig 1A). Specifically, the matured offspring secretes endogenous antibodies with quantitatively similar affinities to bacterial groups as their nursing mother; and relative abundances of bacterial taxa converge to their equilibrium values in host's fecal samples from day 720. We estimate a single set of affinity values shared between the mother and the matured offspring, reflecting maternal antibodies in the gut lumen before activation and after the stabilization of the host's immune response. For this step of inference, we define a composite optimization criterion that integrates the relative and absolute abundances of key taxonomic groups observed in fecal samples of infants and their IgA indexes (indicating enrichment in the SIgA coated (SIgA+) fractions). This optimization step provides estimates that remain fixed during the inference for the interim phase of endogenous

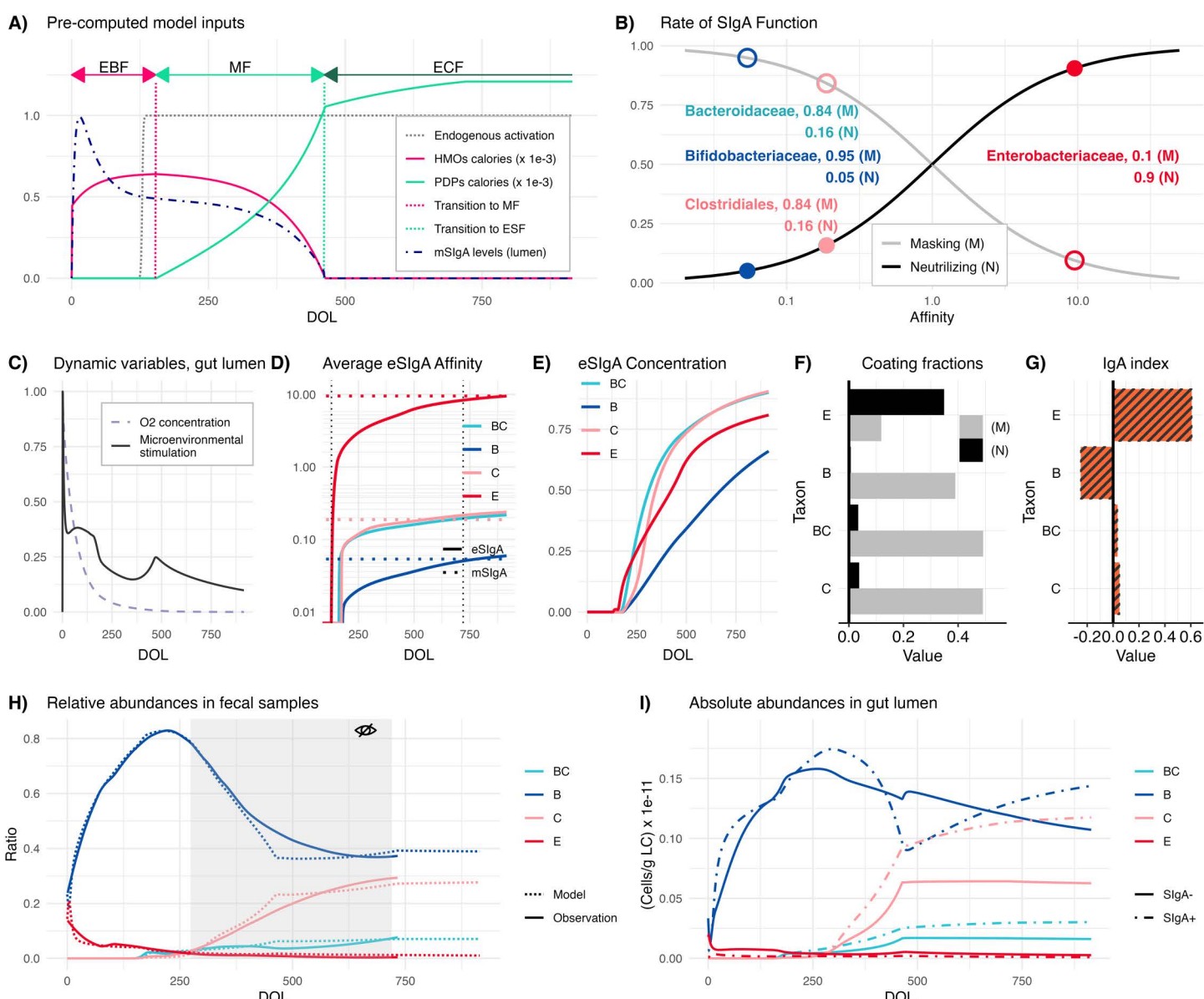

**Fig 2. Illustration of Core Mechanisms and Model Fit. (A)** Model inputs, including normalized maternal secretory Immunoglobulin A (mSIgA) concentration, human milk oligosaccharide (HMOs), and plant-derived polysaccharides (PDPs) caloric inputs, and timing of the endogenous immune system activation over time, noting that the duration of the different forms of feeding can be altered. EBF: exclusive breastfeeding; MF: mixed feeding; ECF: exclusive complementary feeding. **(B)** Rates and types of SIgA function relative to the level of antibody affinity, where mSIgA values for different taxa are highlighted. Filled and unfilled circles represent the rates of neutralizing and masking, respectively. **(C)** Normalized oxygen concentration and microenvironmental stimulation (normalized by its peak value, reflecting the inflammatory tone of the environment) over time. **(D)** Temporal progression of average endogenous SIgA (eSIgA) affinities (solid lines) gradually approaching the mSIgA affinities (dashed lines) and **(E)** the normalized antibody concentrations in the gut lumen for each key taxon. **(F)** Fractions of masked (M) and neutralized (N) bacteria and **(G)** IgA indexes for each taxon at DOL 735. **(H)** Relative abundances in fecal samples aligned with observational data, where the gray area highlights the out-of-sample data during parameter inference. **(I)** Absolute abundances in the gut lumen for key taxa distinguished by coating status. Model fit estimates for all parameters with their respective description, unit, and prior distribution are provided in S3 Table. DOL: Day of life; E: *Enterobacteriaceae*; B: *Bifidobacteriaceae*; BC: *Bacteroidaceae*; C: *Clostridiales*; (M): Masking; (N): Neutralizing. The data underlying this figure can be found in https://doi.org/10.5281/zenodo.15629746.

immune system maturation. A second composite optimization criterion is applied to match the average SIgA coating ratio [36] in the gut lumen and the matured endogenous affinity values to the maternal ones.

To capture immunological and competitive interactions across ontogeny, we selected four bacterial taxonomic groups to reflect broad categories of host-bacterial interactions: *Enterobacteriaceae* encompasses the *Escherichia-Shigella* genus with potentially pathogenic [37,38] bacteria encountered pre-weaning when the endogenous system is more vulnerable to enteric infections. *Bifidobacteriaceae* represents early colonizers with anti-inflammatory properties that regulate the micro-environment and provide colonization resistance. *Bacteroidaceae* and *Clostridiales* are post-weaning bacteria with potentially anti-inflammatory properties that contribute to a balanced gut environment when properly regulated by SIgA (S2 Table). Taxa are differentiated by two parameters: (*i*) their invasiveness [39], defined as their potential to penetrate intestinal epithelial cells, and (*ii*) their immunostimulatory potential, defined as their capacity to modulate the inflammatory tone of the microenvironment [40,41] via their metabolic products and epithelial conditioning [42]. While the relative ranking of these parameters across taxa is established *a priori* based on literature – for example, *Escherichia-Shigella* is assumed to have higher invasiveness and immunostimulatory potential than the rest of the taxa given its pathogenic potential – their specific values are inferred during model fitting (S3 Table). Although bacterial niche (such as proximity to the epithelium [43]) could serve as an additional parameter for differentiating taxa in relation to SIgA responses, it has not been included in the model due to lack of existing quantitative data.

There is substantial evidence that the transfer of maternal antibodies to the infant gut through breastmilk plays a crucial role in shaping the neonatal gut microbiome and the immune system's development [19,44,45]. We characterize SIgA's functionality into two non-exclusive categories: masking (M), which represents SIgA's non-neutralizing functionality, promoting the intestinal residency of the symbiotic commensals by limiting their interactions with the immune system [46,47]; and neutralizing (N), contributing to the expulsion of bacteria from the gut [43]. The masking (M) and neutralizing (N) rates per one unit of SIgA concentration are both modeled as a graduated response to affinity levels [48] (Eq. 1–2) based on the observation that SIgA targeting pathogens or pathobionts generally exhibits higher affinity compared to SIgA targeting symbiotic commensals, which often neutralizes rather than promotes their presence [49]. These rates are scaled by the binding ability of the antibodies, which we define as the strength of association between an antibody and its target antigen based on its affinity (Eq. 3). This dual function of SIgA creates three subpopulations in the gut lumen — uncoated (SIgA-), masked (SIgA+ (M)), and neutralized (SIgA+ (N)) — of which only the uncoated and masked are metabolically active. The neutralized subpopulation neither interacts with other subpopulations nor stimulates the immune system. Because both masked and neutralized populations are coated with SIgA, techniques like IgA-Seq or BugFACS would detect binding in both cases, and the resulting SIgA coating ratio would reflect contributions from both types of bacteria. Furthermore, the functional dichotomy between high-affinity neutralizing and low-affinity masking SIgA is not absolute, as SIgA binding can influence bacterial growth or colonization in species-specific ways, involving mechanisms such as enchained growth, biofilm formation, or altered gene expression [17]. Given the scale of data available, our model does not attempt to capture these nuances (S1 Text, section Model Limitations), but instead focuses on a fundamental, bacteria-agnostic mechanism, aiming to capture the core features of SIgA-microbiome interactions. Rather than relying on species-specific rules, our framework explores how affinity and functionality shift dynamically under different physiological conditions such as homeostasis and inflammation.

Our model assumes a common set of mechanisms for endogenous SIgA maturation across all commensals, allowing rates of antibody masking and neutralization to emerge from maturing affinity levels, as shaped by a backdrop of time-varying processes including developmental stages and dietary patterns (Fig 2A). Our quasi-stochastic formulation of GC reactions assumes a temporally dynamic distribution of B-cell receptor (BCR) binding variability that dictates B cell fates; mean and standard deviation of binding variability are adjusted at each proliferation, somatic hypermutation, and selection (P-SHM-S) cycle (Fig 1C). Naïve B cells migrate to the GALT inductive sites, are activated depending on the antigen uptake by M cells (Eq. 5–7), and start participating in GC reactions [50]. Each P-SHM-S cycle is informed by the

microenvironmental stimulation in the PPs, and the selection pressure imposed by an implicit model of antigen-specific follicular helper and regulatory T cells (Materials and methods, Modeling the role of germinal centers, and Eq. 29), eventually determining the size and average affinity of the plasma cell pool. Microenvironmental stimulation is modeled as a composite variable (Eq. 8) reflecting the cumulative effects of cytokines and microbial products; akin to the inflammatory tone of the local environment [51]. We infer mSIgA affinities (Fig 2B) assuming the nursing mother with a balanced microbiome-immune response transfers masking antibodies for the symbiotic commensals (*Bifidobacteriaceae*, *Bacteroidaceae*, and *Clostridiales*); and neutralizing antibodies for the pathogenic taxon *Enterobacteriaceae* [48,52–56]. Model fit estimates illustrate the trajectory of microenvironmental stimulation (Fig 2C), average affinity of the endogenous antibodies converging toward maternal values (Fig 2D), along with the normalized concentrations of antibody levels (Fig 2E) proportional to the maturation of plasma cells being imprinted as the host ages.

Bacterial colonization is also affected by the oxygen concentration ($O_2$) in the gut lumen (Eq. 4). $O_2$ inhibits the growth of obligate anaerobic bacteria and is consumed by facultative anaerobes (Eq. 24) [57]. From an initially normalized level of 1, $O_2$ degrades at a rate inferred during the model fitting process (Fig 2C).

To parametrize our model, we use relative bacterial abundance and IgA-Seq data derived from infants' fecal samples [20–23] as a proxy for the ecological and immunological dynamics within the gut lumen [58]; however, equating fecal samples with the gut lumen overlooks SIgA's role in microbial turnover [46,47]. We reconstruct the relative abundances per fecal content by calculating the weighted sum of neutralized, coated, and uncoated subpopulations in the gut lumen. This calculation incorporates subpopulation-specific observation rates (Eq. 14–17), which represent the proportion of each bacterial subpopulation that successfully transfers from the gut lumen to fecal samples. These observation rates are inferred during the model fitting process and differ between neutralized, coated, and uncoated bacteria (S3 Table), accounting for their varying propensities to appear in fecal samples relative to their actual abundance in the gut lumen. This approach translates the true microbial abundances in the gut lumen to the measurable abundances detected in fecal samples (Fig 1A), allowing us to align the model output with the 16S rRNA sequencing analysis provided in *Tsukuda and colleagues* [20]. We apply the same reconstruction using SIgA−, SIgA+(C), and SIgA+(N) subpopulations to compute the IgA indexes for each taxon, allowing us to include the IgA-seq profiles of *Enterobacteriaceae* and *Bifidobacteriaceae* provided in *Planer and colleagues* [21]. Our estimates indicate an average coating ratio of 48.8%, with a negative IgA index for *Bifidobacteriaceae*, slightly positive IgA indexes for *Bacteroidaceae* and *Clostridiales*, and a positive IgA index for *Enterobacteriaceae* (Fig 2F).

Relative abundance estimates for periods of inactivity (neonatal, early postnatal) or stability (after 2 years) of the endogenous immune system demonstrate the performance of our model fitting procedure (Fig 2G). Furthermore, out-of-sample data corresponding to the period of maturation—not used for parameter inference (Fig 2G, gray area)—was also accurately predicted by the model. This alignment provides indirect support of the underlying assumptions regarding the immune response's impact on microbial community dynamics, lending credibility to the model estimates of the gut lumen (Fig 2H) for which direct data is unavailable (Fig 1A).

Relative abundance estimates aligned with data points are depicted in S1 Fig, and additional assumptions implicit to the model structure are provided in S3 Table. A shiny app is available, to allow exploration of model parameters and emergent phenomena.

## IgA-bound Enterobacteriaceae abundance can predict immune phenotypes but only at specific ontogenetic phases

Our ability to develop effective intervention strategies for preventing pathological imprinting hinges on our capacity to accurately predict the trajectory of 'immune education' as the infant gut matures. While there are studies correlating the time course of microbiome composition and IgA-binding patterns with eventual disease susceptibility [59–61], the predictive potential of such analyses at different stages of ontogeny remains underexplored. To evaluate the potential of using

taxonomic and IgA-seq profiles of fecal samples to predict the trajectory of infant's immune maturation, we systematically explored variation in ecological and immunological outcomes by simulating our model for a total duration of 735 days across a comprehensive range of exclusive breastfeeding (EBF) and mixed feeding (MF) durations. Across 75.2% of all feeding scenarios, the host's gut mucosal immune response matured to a tolerant profile by simultaneously developing predominantly masking SIgA (rate of masking being above 0.5, higher than rate of neutralizing) against all symbiotic commensals (*Bifidobacteriaceae*, *Bacteroidaceae*, and *Clostridiales*) (Fig 3A), although the underlying rate of masking varied (Fig 3B). The infant immune response diverges to a hyperreactive profile—developing predominantly neutralizing antibodies against at least one of the symbiotic commensals across 24.8% of all possible feeding scenarios (Fig 3A). These results suggest robust convergence towards a tolerogenic profile across a wide spectrum of feeding patterns, in line with previously reported stereotypical convergence of the systemic immune response [2].

Using this synthetically generated data, we explored the diagnostic potential of early life fecal microbiome data in predicting emergent immunopathology. Specifically, we analyzed how effectively the total and SIgA-bound *Enterobacteriaceae* abundance from fecal samples collected at various developmental stages could discriminate between individuals who ultimately developed tolerance versus hyperreactivity after 2 years (at DOL 735). We focused on SIgA-bound Enterobacteriaceae as our primary marker based on the evidence that SIgA coating patterns can identify bacteria associated with immune dysregulation and inflammatory conditions [43]. We first compared the total and SIgA-bound *Enterobacteriaceae* abundance between tolerant and hyperreactive subjects at different time points throughout development. Higher levels of SIgA-bound *Enterobacteriaceae* abundance gradually flip from reflecting tolerant to reflecting hyperreactive immunological states over the time course of ontogeny (Fig 3D). Lower SIgA-bound *Enterobacteriaceae* observed for hyperreactive phenotypes during early phases of ontogeny are indicative of insufficient *Enterobacteriaceae* neutralization by maternal antibodies. As the host matures, this pattern alters, with the hyperreactive phenotype exhibiting an increase in SIgA-bound *Enterobacteriaceae* abundance relative to the tolerant state, echoing the enrichment in SIgA-coated *Enterobacteriaceae* in adults with IBD-like phenotypes [62,63].

Building on these associations, we next asked how well total and SIgA-bound *Enterobacteriaceae* abundances at different stages of ontogeny could predict immunological phenotypes. We implemented a 5-fold cross-validation approach with stratified sampling to account for potential class imbalances in our binary outcome. For each time point, we fitted three logistic regression models including as covariates: (i) total *Enterobacteriaceae* abundance, (ii) SIgA-bound *Enterobacteriaceae* abundance, and (iii) both total and SIgA-bound *Enterobacteriaceae* abundances. To assess and compare the predictive performance of each model, we calculated the area under the receiver operating characteristic curve (AUC) along with standard errors at each time point. This metric quantifies how well each model discriminates between the two outcome categories, with higher AUC values indicating better classification ability.

Total *Enterobacteriaceae* abundances had the lowest predictive capacity during the first 6 months of life (Figs 3C and 2E), which can be explained by considering the two compensatory mechanisms regulating it: (*i*) ecological competition preventing pathogen overgrowth and (*ii*) early activation of the endogenous immune system (earliest at DOL 30) when maternal antibodies are not sufficiently neutralizing. In contrast, SIgA-bound *Enterobacteriaceae* abundance alone during the very early days of life provides valuable information regarding the immunological trajectory (Fig 3D and 3E), as this quantity reflects the efficiency and/or presence of maternal antibodies [60]. Starting from month 6 (DOL 180), the predictive power of total *Enterobacteriaceae* abundance increases while that of SIgA-bound *Enterobacteriaceae* decreases. This shift indicates that endogenous immune responses against symbiotic commensals combined with ecological competition become the primary regulators of *Enterobacteriaceae* population, exerting stronger selection pressure than the endogenous SIgA response to *Enterobacteriaceae* itself (S2 Fig). This shift in predictive powers also coincides with the transition from maternal to endogenous antibody responses. Notably, SIgA-bound *Enterobacteriaceae* samples collected during this transitional period from maternal to endogenous SIgA dominance in the gut lumen provided the least information about the host's immunological trajectory, revealing non-monotonic temporal patterns in the predictive value of 16S rRNA and

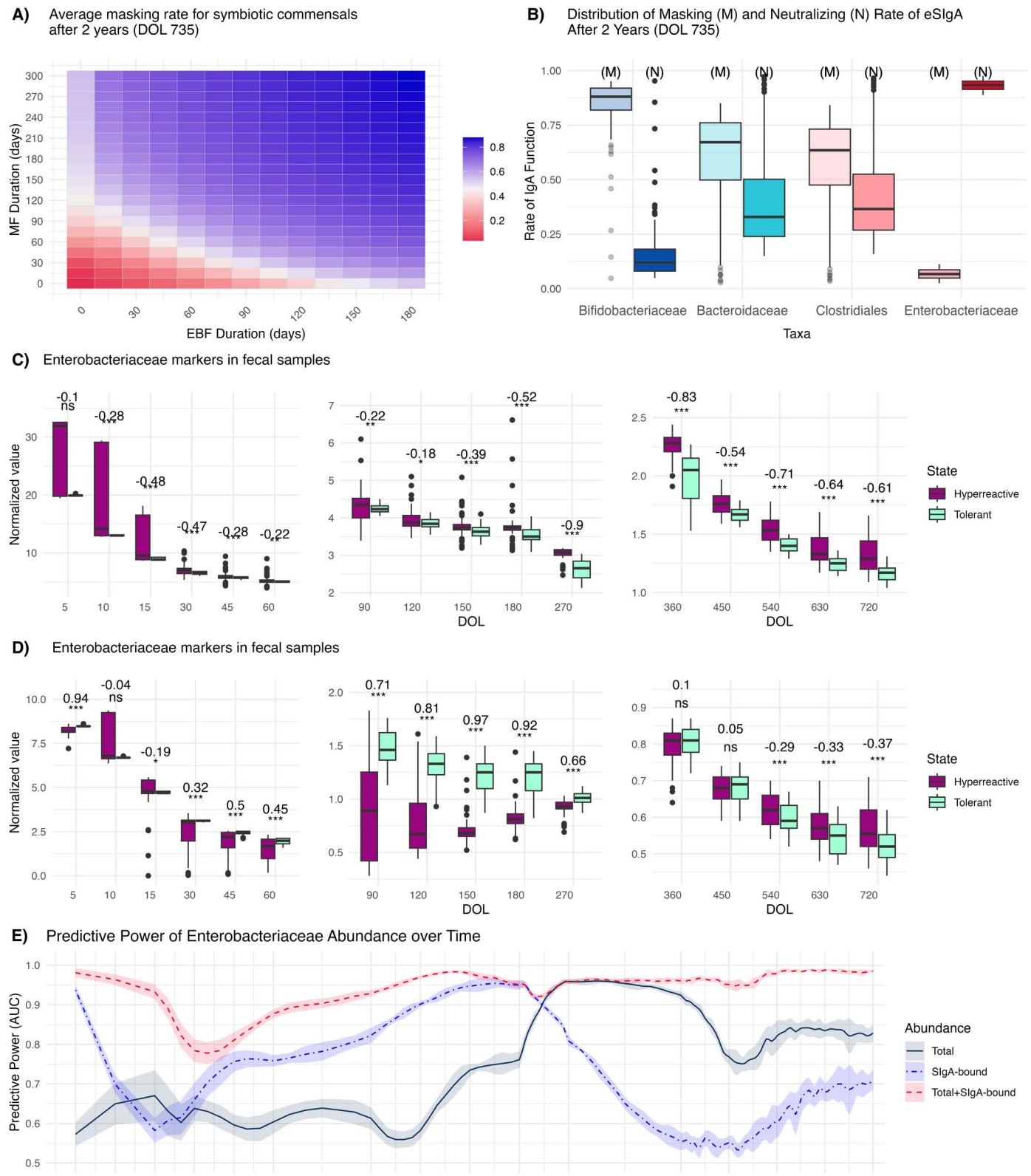

**Fig 3. Immunological outcomes and *Enterobacteriaceae* abundances for the first 2 years of life for different combinations of exclusive breastfeeding (EBF) and mixed feeding (MF) durations. (A)** Immune functionality across a spectrum of EBF and MF durations, where 'tolerant'

corresponds to developing predominantly non-neutralizing but masking antibodies against all symbiotic commensals; and 'hyperreactive' otherwise. EBF and MF durations vary from 0 to 180 and to 300 days, respectively **(B)** Distribution of masking (M) and neutralizing (N) rates of endogenous SIgA at DOL 735. Distribution of (C) total and **(D)** SIgA-bound *Enterobacteriaceae* abundances over the first two years of life segregated by tolerant (predominantly masking SIgA for all symbiotic commensals simultaneously) and hyperreactive (predominantly neutralizing SIgA for at least one of the symbiotic commensals) states. The dataset is balanced for each time point to reflect the typical sample size of recent cohort studies [65]. A Wilcoxon rank-sum test is used to calculate the effect size (rank-biserial correlation from −1 to +1; values closer to ±1 indicating stronger associations) and the significance (*** for $p < 0.001$, ** for $p < 0.01$, * for $p < 0.05$,. for $p < 0.1$, and ns for $p \geq 0.1$), displayed at the top of each boxplot. **(E)** Mean predictive power (AUC) of *Enterobacteriaceae* abundance segregated by data types. Predictive power of 0.5 indicates no better prediction of a tolerogenic/hyperreactive phenotype than random chance; 1 indicates perfect accuracy. Shaded areas represent ±1 standard error around the mean values. Values are smoothed using a local polynomial regression (LOESS) model. DOL: Day of life; AUC: Area under the receiver operating characteristic curve. The data underlying this figure can be found in https://doi.org/10.5281/zenodo.15629746.

IgA-seq analyses. Comparison with an independent dataset provided in [64] showed that our model captured similar patterns in the relative importance of microbial and inflammatory predictors (S1 Text, section Illustrative validation using an external dataset, S12 Fig), providing additional support to the plausibility of our results.

### Ecology versus the microenvironment: what carries the information?

An enduring mystery in mucosal immunology is how postnatal influences on the immune system persist into adulthood, even though many core cell types are not present when these initial influences occur [11] (S1 Text, section Role of Early Life Infections, S3 Fig). One way to test whether this persistent influence is also reflected in our model is to investigate the impact of EBF duration on affinity maturation: antigenic sampling is delayed if M cells remain immature as a result of persistently high levels of mSIgA, and this postpones the activation of B and T cells until the end of this period. Thus, we expect the influence of the EBF period—if any—to rapidly fade unless other components of the system have enduring effects, since mSIgA has a strict half-life and the delivery of HMOs ceases instantaneously with the cessation of breastfeeding (S4A Fig).

We first investigate the differential impacts of EBF and MF durations in determining endogenous SIgA (eSIgA) affinity across a comprehensive range of EBF and MF durations. Given the presence of pre-weaning commensals (*Bifidobacteriaceae*, *Enterobacteriaceae*) in the gut lumen during both EBF and MF, eSIgA affinity maturation towards these taxa will be influenced by the duration of both feeding practices. Conversely, a significant role for EBF duration in the affinity maturation against post-weaning commensals (*Bacteroidaceae*, *Clostridiales*) would align with the persistent postnatal influences described above. As hypothesized, immune responses against pre-weaning and post-weaning commensals are impacted by EBF and MF durations differently shown by a predictor importance analysis (Fig 4A), where we compare the relative importance of EBF and MF durations in explaining the variance in eSIgA affinities. Recapitulating the presented conundrum, the influence of EBF duration remains prominent in explaining the variance in eSIgA affinities against post-weaning commensals, despite MF duration being ~1.5 times more influential. This influence is more prominent when EBF is followed by Exclusive Complementary Feeding (ECF) without any intervening MF period, which makes it challenging for the host to develop tolerance against post-weaning commensals (S4B Fig).

Although mSIgA itself may not persist, its initial modifications may continue to shape both the ecology and the inflammatory tone of the microenvironment; similarly, HMOs may have shaped the ecology by selecting for bacteria based on their metabolic capacity, leading to a sustained temporal influence. Therefore, we compare the impact of pre-weaning commensal abundance and the microenvironmental stimulation on eSIgA affinities for post-weaning commensals to identify which components of our system carry information from the EBF period onward. We employed the Random Forests method to handle correlation between microbial abundances and microenvironmental stimulation. Intriguingly, our findings reveal a significantly greater influence of the microenvironmental stimulation over taxonomic composition (Fig 4B). These findings suggest that the conditions and signals within the gut microenvironment—such as epithelial permeability and the

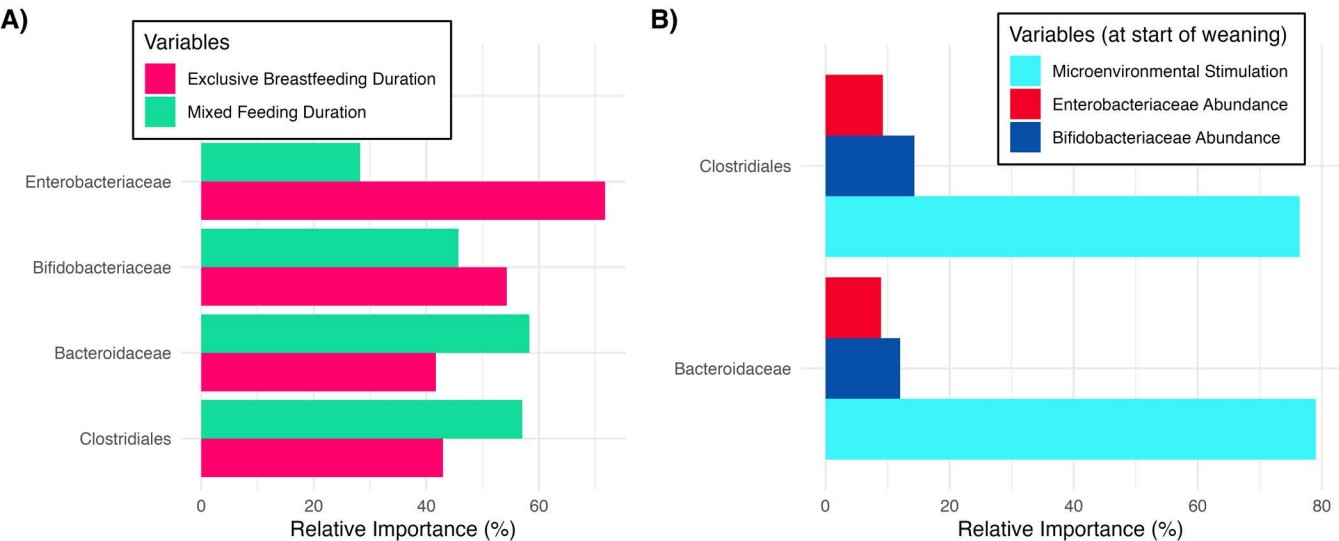

**Fig 4. Impact of different feeding durations and environmental variables in determining the matured eSIgA values for key taxa. (A)** Relative importance analysis in determining the eSIgA values for key taxa, comparing the impact of EBF and MF durations. **(B)** Relative importance analysis in determining the eSIgA values for post-weaning commensals, comparing the impact of the abundances of pre-weaning commensals and the microenvironmental stimulation. The data underlying this figure can be found in https://doi.org/10.5281/zenodo.15629746.

tolerogenic bias of follicular T cells which are directly modulated by the microenvironmental stimulation in our model—play a pivotal role in shaping the infant's immune response relative to the presence or abundance of specific bacterial taxa. Moreover, these results indicate that breastfeeding's influence extends beyond immediate transmission of HMOs, antibodies, and microbes, conditioning the gut microenvironment in ways that produce effects outlasting the transient presence of such factors. While inflammatory tone and immune maturation undoubtedly have bidirectional interactions in our model making causality difficult to disentangle, our analysis provides evidence for directionality through temporal precedence: early microenvironmental conditions at the start of weaning predict subsequent immune responses against taxa that only appear post-weaning. This temporal sequence helps untangle the cyclic relationship and supports the directionality from early inflammatory tone to downstream immune maturation.

### Maternal protection and immunological imprinting in the offspring depends on mother's immunological profile

Despite the well-established health benefits of breastfeeding, it presents a complex decision-making process for mothers experiencing immune dysregulation [66]. While epigenetic regulation of various immune cells, including dendritic cells (DCs), stromal cells, T cells, alveolar macrophages, and epithelial cells, has been implicated in the persistence of transgenerational immune priming [12,59], other mechanisms may also play a role. By meticulously eliminating other possibilities such as genetic and epigenetic factors, microbiota variation, and differences in milk-derived metabolites, recent work by Ramanan *and colleagues* demonstrated that the primary maternal factor allowing for transgenerational immune priming in mice is vertical transmission of maternal IgA [11]. This observation opens up the possibility of transmission of inflammatory phenotypes via maternal IgA, as well as tolerogenic ones. Indeed, evidence suggests that IBD is significantly more prevalent in the offspring of IBD mothers, who tend to produce breastmilk with higher inflammatory potential and lower antibody levels [67–69]; and inadequate levels of maternal SIgA to food allergens have been linked to infantile allergic diseases in breastfed infants, showing a stronger correlation with symptoms than parental atopic history [70]. However, breastmilk's influence goes beyond immune priming and antibody transfer, also enhancing colonization resistance against

pathogens through nutritional and microbial support. Therefore, it is not immediately apparent under what circumstances the benefits of breastfeeding outweigh its risk of inflammatory priming.

To address this question, we compare five different feeding scenarios with varying maternal immunological profiles: (*i*) control group recapitulating the scenario in Fig 2, (*ii*) excessive targeting of commensal bacteria by maternal antibodies (high affinity mSIgA against *Bifidobacteriaceae*, *Bacteroidaceae*, and *Clostridiales*), mimicking an IBD-like phenotype [71], (*iii*) breastmilk devoid of mIgA but supplying HMOs and *Bifidobacteriaceae*, indicative of a maternal SIgA deficiency, (*iv*) ECF, with *Bifidobacteriaceae* introduced at a lower inoculum size and delayed introduction of post-weaning taxa (*Bacteroidaceae* and *Clostridiales*), mimicking a scenario with no breastfeeding, and (*v*) ECF with *Bacteroidaceae* and *Clostridiales* transfer from the start, mimicking additional probiotic supplementation. In all scenarios, *Enterobacteriaceae* are introduced at the same inoculum size to assess the balance between pathogenic and symbiotic commensals (*Bifidobacteriaceae*, *Bacteroidaceae*, and *Clostridiales*) during the first 30 days of life, a critical period when the infant is particularly vulnerable to pathogen overgrowth.

Scenarios differed in degree of pathogen control, demonstrating two opposite trends (Fig 5). ECF with probiotic supplementation (Fig 5A and 5C; dashed dark purple line) and SIgA-deficient breastfeeding (Fig 5A and 5C; dashed yellow line) initially showed an increase in the pathogenic-to-symbiotic commensal ratio followed by a decrease that brought them to a similar level by day 10, echoing their similarity in leading to Necrotizing enterocolitis [60] and sepsis susceptibility [72]. SIgA-deficient breastfeeding had an initial advantage suggesting that HMOs and *Bifidobacteriaceae* can partially compensate for the lack of mSIgA by directly modulating the gut microbiome composition and thus the colonization resistance against *Enterobacteriaceae*. In contrast, ECF without probiotic supplementation (Fig 5A and 5C; light green line) resulted in the highest pathogenic-to-symbiotic commensal ratio across all scenarios, emphasizing the importance of key symbiotic commensals in providing defense against pathogens. Interestingly, the second worst outcome in terms of pathogen control was observed with the IBD-like phenotype (Fig 5 dotted magenta line), even though it provides neutralizing antibodies against the pathogenic commensal *Enterobacteriaceae*. The transfer of hyperreactive mSIgA against symbiotic commensals allowed for excessive *Enterobacteriaceae* overgrowth by neutralizing *Bifidobacteriaceae*, leading to significantly higher relative abundances of SIgA-bound *Enterobacteriaceae* in the fecal samples compared to the control group (Fig 5B).

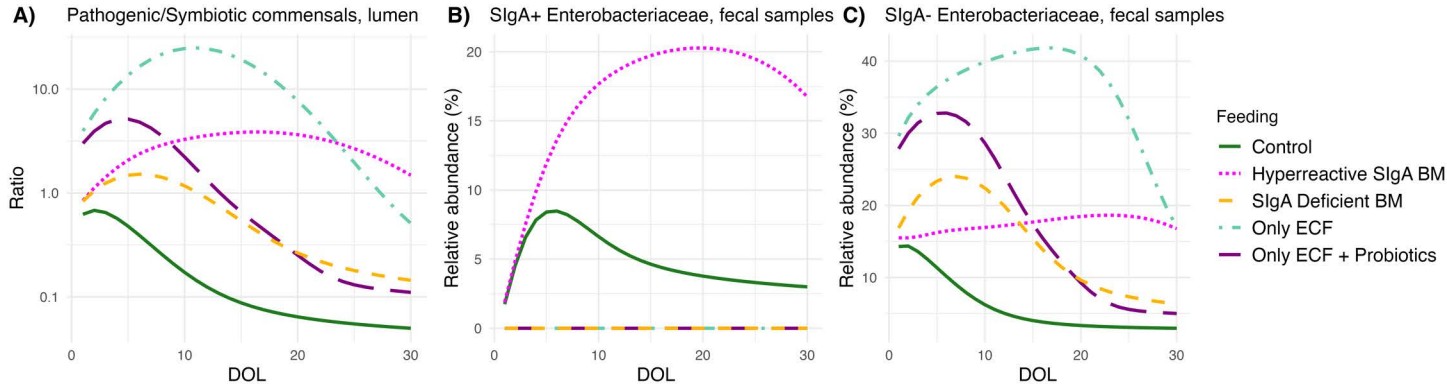

**Fig 5. Comparison of various breastfeeding scenarios and their impact on *Enterobacteriaceae* growth.** Temporal progression of **(A)** Pathogenic-to-symbiotic commensal ratio in the gut lumen, **(B)** relative SIgA-bound (SIgA+) *Enterobacteriaceae* abundance in fecal samples, and **(C)** relative SIgA-free (SIgA−) *Enterobacteriaceae* abundance in fecal samples of the infant for the first 30 days of life (DOL) for four different breastfeeding scenarios: Control, breastmilk (BM) with hyperreactive mSIgA, mSIgA deficient BM, and only exclusive complementary feeding (ECF). The data underlying this figure can be found in https://doi.org/10.5281/zenodo.15629746.

In all scenarios where breastmilk lacked mSIgA, we observed hyperreactive eSIgA maturation in the infant, highlighting the influence of breastmilk antibody levels in immune imprinting (Table 1 and S5 Fig), consistent with findings in the literature [67–70]. In terms of the transgenerational transmission of IgA affinity profiles, our model demonstrates a modulation of hyperreactivity against symbiotic commensals rather than complete transfer of maternal profiles (Table 1 and S5 Fig). Offspring hyperreactivity against symbiotic commensals is attenuated relative to maternal hyperreactivity through both antigen-specific and antigen-agnostic mechanisms. In the antigen-specific pathway, the invasiveness of individual bacterial taxa influences offspring developing immune responses, modulated by the limited masking capacity of high-affinity mSIgA in breastmilk (S8 Fig). This limited mSIgA masking capacity, combined with the M cells' selective bias for sampling IgA-bacteria complexes [73,74] promotes the accumulation of IgA-coated bacterial antigens in the GALT inductive sites, which activates tolerogenic DCs [75] and dampens the development of high-affinity neutralizing antibodies against symbiotic commensals (Fig 6B–6D). Simultaneously, an antigen-agnostic pathway operates through microenvironmental stimulation, driven by the cumulative inflammatory capacity of unmasked bacterial subpopulations, with mSIgA's limited masking capacity contributing to a minor reduction in inflammation. Unlike symbiotic commensals, pathogenic bacteria are inherently more invasive, making the mSIgA affinity and the IgA-mediated M cell sampling bias less influential for the taxonomic group *Enterobacteriaceae* in our model (Fig 6A). The complete transmission of tolerance and the attenuated transmission of hyperreactivity suggest an overall beneficial effect of breastfeeding for developing tolerance against symbiotic commensals, even when the mother has hyperreactive mSIgA in her breastmilk which leads to a suboptimal endogenous antibody development (Table 1, row "Hyperreactive SIgA BM"). This attenuated transmission likely provides a mechanism for gradual normalization of immune responses across generations. However, it is important to note that this result of attenuated transmission of hyperreactivity does not account for potential contributions from maternally derived cytokines (e.g., IL-1β, IL-6, TNF-α, TGF-β) or other mediators transferred through the placenta or breastmilk [76].

**Table 1. SIgA profiles in the mother and the offspring for various breastfeeding scenarios.**

| | | Neutralizing (N) rates of SIgA for each taxon | | | |
| | | *Enterobacteriaceae* | *Bifidobacteriaceae* | *Bacteroidaceae* | *Clostridiales* |
|---|---|---|---|---|---|
| Scenario | Host | | | | |
| Control | Mother | 0.90 | 0.05 | 0.16 | 0.16 |
| Control | Offspring | 0.89 | 0.05 | 0.17 | 0.18 |
| Hyperreactive SIgA BM | Mother | 0.90 | 0.90 | 0.90 | 0.90 |
| Hyperreactive SIgA BM | Offspring | 0.92 | 0.16 | 0.54 | 0.53 |
| SIgA Deficient BM | Mother | – | – | – | – |
| SIgA Deficient BM | Offspring | 0.98 | 0.97 | 0.97 | 0.97 |
| Only ECF | Mother | – | – | – | – |
| Only ECF | Offspring | 0.98 | 0.98 | 0.96 | 0.88 |
| Only ECF + Probiotics | Mother | – | – | – | – |
| Only ECF + Probiotics | Offspring | 0.98 | 0.95 | 0.97 | 0.97 |

Each column shows the neutralizing (N) rates of SIgA for each taxon, scenario, and host (mother or offspring) combination, as analyzed in Fig 5. Cell colors indicate the magnitude of the neutralizing (N) rates on a blue-to-red gradient—red representing higher values (with darker red indicating the highest), blue representing lower values, and white indicating zero—aligned with the color scheme in Fig 6. Note that entries are left blank for scenarios in which the mother either does not breastfeed or has IgA deficiency. Rows for the scenario "Hyperreactive SIgA BM" demonstrate the attenuated transmission of SIgA hyperreactivity from the mother to the offspring. The level of maternal SIgA hyperreactivity is attenuated 82% for *Bifidobacteriaceae* (maternal neutralizing rate of 0.9 translates to a neutralizing rate of 0.16 in the offspring), and 40% for *Bacteroidaceae* and *Clostridiales* (maternal neutralizing rate of 0.9 translates to neutralizing rates of 0.54 and 0.53 in the offspring, respectively). The data underlying this table can be found in https://doi.org/10.5281/zenodo.15629746.

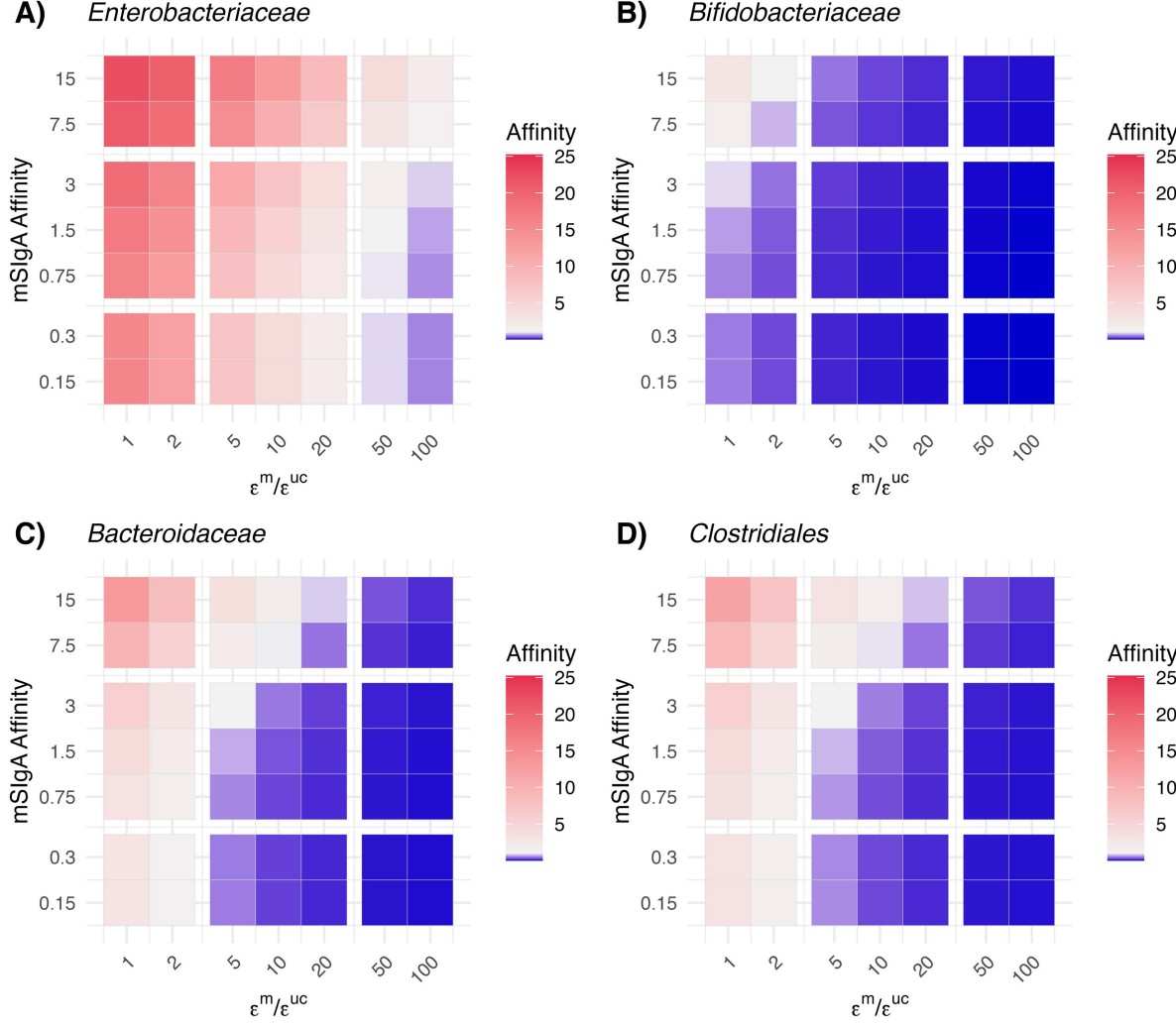

**Fig 6. Combined influence of maternal SIgA affinity and antigenic sampling rates on the endogenous SIgA affinities.** Heatmaps show endogenous SIgA (eSIgA) affinity levels at two years (DOL 735) across combinations of maternal SIgA (mSIgA) affinities and the ratio of M-cell antigenic sampling rates for masked vs. uncoated bacterial antigens (represented by $\in^m$ / $\in^{uc}$, where $\in^m$ and $\in^{uc}$ are the antigenic-sampling rate of masked and uncoated bacterial antigens by M cells, respectively) for **A)** *Enterobacteriaceae*, **B)** *Bifidobacteriaceae*, **C)** *Bacteroidaceae*, and **D)** *Clostridiales*. Color gradients reflect the magnitude of eSIgA affinity: red indicates values greater than 1 (predominantly neutralizing eSIgA), white corresponds to 1, and blue indicates values below 1 (predominantly masking eSIgA). The data underlying this figure can be found in https://doi.org/10.5281/zenodo.15629746.

## Discussion

'Immune education' has proved a powerful concept in understanding the long shadow of early life on health outcomes. Yet, to date, the concept has remained largely qualitative, informally encompassing a wide range of individual components. This diversity and complexity of interactions make integrating these effects by experimental methods alone intractable, quite apart from the question of the degree to which animal models translate to humans. We take the first steps towards formalizing the concept of immune education in the context of gut mucosal SIgA–microbiome interactions, leveraging the power of mathematical frameworks to robustly integrate known feedback loops connecting the ecology and immunology of the gut, and inferring core parameters using data on early life immunity and microbial community dynamics.

Our analysis offers novel qualitative insights that, while grounded in our mechanistic formulations and model assumptions, may inform clinically relevant hypotheses. First, it adds a predictive dimension to the interpretation of gut microbial composition data. Traditional studies, employing 16S rRNA and IgA-seq techniques on fecal samples, have primarily focused on identifying associations between early-life microbiome compositions and later immune-mediated health outcomes. We add mechanistic understanding as to why and how the IgA coating patterns vary over the course of ontogeny by disentangling the influences of maternal and endogenous antibodies. We show that IgA-seq analysis of fecal samples collected during the transitional period—from the gradual decline of maternal antibodies to the rise of endogenous SIgA—has limited predictive value for immunological outcomes. We also reveal opposing temporal trends: reduced SIgA-bound *Enterobacteriaceae* in early ontogeny indicates insufficient maternal neutralization and predicts hyperreactivity, while the same pattern later suggests tolerogenic imprinting. These non-monotonic temporal patterns suggest that the timing of sample collection critically shapes the ability of 16S rRNA and IgA-seq data to predict long-term immune phenotypes. Moreover, they emphasize the importance of timing in study design and clinical sampling strategies, particularly when sample availability or resources are limited. These findings can be empirically tested through longitudinal studies that assess whether the predictive power of IgA-seq varies systematically across ontogeny, potentially revealing optimal sampling windows for forecasting immune-mediated conditions.

MF duration emerges as a critical determinant of the infant's immune trajectory: the sustained presence of maternal antibodies after weaning enables the introduction of novel antigens to the immune system in a non-inflammatory manner, promoting the development of tolerance. EBF continues to shape the immune response against microbes introduced even after its cessation by its remnant effects on the inflammatory tone within the gut lumen and GALT inductive sites. This emergent phenomenon suggests a resolution to the perennial question of how information from the early postnatal phase propagates to further stages of immune development: our model indicates that the information is not specifically carried by either symbiotic or pathogenic taxa, but, rather, results from the cumulative capacity of their SIgA-coated and uncoated subpopulations to modulate the inflammatory tone of the microenvironment.

Beyond these fundamental insights, our analysis systematically explores the applied question of how to approach breastfeeding when maternal immunological conditions such as allergies, IBD, or SIgA deficiency affect breastmilk composition. Both lack of SIgA in breastmilk or excessive targeting of symbiotic commensals by SIgA can lead to suboptimal immune imprinting in the infant, indicating that maternal SIgA is one of the key mechanisms driving the transgenerational transmission of immunological phenotypes. The absence of breastmilk SIgA leads to a fully hyperreactive endogenous SIgA maturation, while hyperreactive maternal SIgA results in an attenuated rather than complete transmission of the inflammatory phenotype. This attenuation occurs through the combination of limited masking by high-affinity maternal SIgA, selective bias of M cells for sampling IgA-bacteria complexes, and the inherently low invasiveness of commensal microbes. Although transfer of *Bifidobacteriaceae* and HMOs cannot entirely compensate for maternal SIgA's role in tolerogenic imprinting, it can partially compensate for the absence of SIgA in pathogen control.

Our findings further provide an evolutionary perspective on the maternal contribution to offspring immune development through breastfeeding. While the absence of breastmilk SIgA leads to significantly impaired immune regulation, our model demonstrates that even "suboptimal" maternal SIgA profiles confer substantial benefits compared to no breastfeeding. This highlights the dual functionality of SIgA: first as an antigen-specific mediator of tolerance, and second as an antigen-agnostic molecule that promotes M cell-mediated antigen sampling [73,74] and programs DCs toward tolerogenic phenotypes [75]. When both functions operate optimally, complete transgenerational transfer of tolerance occurs. However, even when antigen-specific tolerance is impaired (as in maternal inflammatory conditions), the antigen-agnostic properties remain effective, resulting in attenuated rather than complete transmission of maternal hyperreactivity. This mechanism allows for the gradual dilution of inflammatory phenotypes across generations, potentially explaining why some inflammatory conditions do not follow simple vertical transmission patterns despite breastfeeding. Such a system provides evolutionary resilience, allowing for recovery from dysregulated immune states over generational timescales rather than perpetuating inflammatory phenotypes indefinitely.

These findings, combined with the influence of excess microenvironmental stimulation on endogenous SIgA affinity maturation, suggest the possibility of alternative therapeutic approaches for promoting tolerogenic imprinting when maternal immunity is hyperreactive or insufficient. Interventions could focus on strategies to modulate gut inflammation beyond probiotic supplementation, for example, targeting Toll-like receptors such as TLR4, which has been proposed to tackle preterm birth and fetal inflammatory injury [77]. Preliminary results from our model indicate the potential effectiveness of this intervention (S6 Fig), suggesting that tolerance could be induced by reducing the inflammatory potential of gram-negative commensals, similar to the effects observed with TLR4 antagonists. However, a complete exploration of such a scenario would necessitate expanding our model to explicitly incorporate cellular immunity, and to account for the distinct immunological effects of structurally diverse LPS molecules on early-life immune education [78] (S1 Text, section *Model Limitations*).

Dysregulated immune development in children has repeatedly been shown to mediate multiple pathologies, ranging from developmental deficits in low-resource settings to immune-mediated disorders in developed countries [79]. Given the widespread burden of these conditions, developing a holistic mechanistic understanding of the concept of 'immune education' holds tremendous potential for clinical translation. Therefore, the primary motivation and contribution of this work is to establish the foundations of a mathematical framework for mechanistic thinking, generating clear and testable hypotheses for empirical investigation. Given the immense complexity and dynamism of early life, our model is necessarily a simplification, tailored to the range of data currently available, and excluding factors such as innate immune mechanisms, changes in gut permeability, and strain-specific shifts in the gut microbiome. Although quantitative predictions may change, qualitative conclusions are likely to be robust to these features (S1 Text, section *Model Limitations*). While we included an illustrative validation using an external dataset (S1 Text, section Illustrative validation using an external dataset), this serves primarily as a proof of concept. Future clinical, experimental, and measurement approaches are likely to open the way to evaluating our underlying mechanistic assumptions (S1 Text, section Experimental Model Validation). Ultimately, our quantitative analysis offers theoretical insights into pressing questions in the field, identifies the gaps in data collection needed to advance mechanistic understanding, and provides a systematic starting point for future data-driven mathematical models of early-life immune development.

## Materials and methods

### Model description

Our model describes the reciprocal ontogeny of the gut microbiome and the gut mucosal secretory IgA (SIgA) response during the first two years of life, combining the ecology of the gut microbiome with the kinetics of the B cell responses in the gut-associated lymphoid tissues (GALT) and their bidirectional relationship. This period of early life is particularly dynamic, impacted by various external and internal factors such as feeding mode, microbial exposures, and the developmental clock of the host [80]. Given the varying influence of these factors at different stages, we categorize the developmental timeline into three distinct phases:

**Maternal Phase:** This phase represents the time frame from birth to transition to MF, during which the infant is exclusively breastfed and thus the infant gut is primarily influenced by maternal factors. In breastfed infants, maternal secretory IgA (mSIgA) from breastmilk is active in the gut lumen, and nourishment primarily comes from human milk oligosaccharides (HMOs). Key immune structures like GCs and M cells in the GALT inductive sites (Peyer's Patches (PPs) in the small intestine and the colonic patches in the colon) are not fully developed until decreasing concentrations of the maternal factors in the breastmilk favor M cell maturation, delaying the onset of an endogenous secretory IgA (eSIgA) response [35]. Naïve B cells migrate to gut lymphoid tissues but remain inactive until M-cell-mediated antigenic sampling begins [81].

**Developmental Phase:** This phase encompasses the transition from EBF to MF and continues until the microbiome composition and eSIgA response stabilize, including the shift to exclusive complementary feeding (ECF). As the influence of maternal antibodies diminishes, the infant's diet increasingly includes PDPs instead of HMOs, meeting the total caloric

need of the infant. Introduction of PDPs increases the overall taxonomic diversity of the gut microbiome by introducing novel antigens. Concurrently, GCs and M cells mature, fostering the development of the eSIgA response, a phenomenon often referred to as the "weaning reaction" [10]. Naïve immune cells become activated, participate in GC reactions, and eventually produce plasma cells secreting IgA.

**Steady Phase:** This is the final phase where the host achieves a stable, complex microbiome and a fully matured endogenous adaptive immune system which, crucially, resembles their mother's SIgA response [21]. The average affinity of the plasma cell pool and the relative abundance of key microbial populations reaches a steady state, indicative of a homeostatic gut ecosystem. During this phase, the host's diet is based solely on PDPs.

While the same equations apply across all phases, separating them into maternal, developmental, and the steady phases is crucial for parameter inference, which will be elucidated in the subsequent sections.

Our mathematical model consists of a system of ordinary and delay differential equations tracking bacterial abundances in the gut lumen and the fecal samples of the host, as well as the immune response focusing on antigen-specific SIgA to gut commensals and their affinity maturation process. A compartmental diagram illustrating the dynamics of the model is given in Fig 1 (see Introduction), model parameters with their corresponding notations, descriptions, units, and numerical ranges are given in S3 Table, and the set of differential equations governing our system are provided via Eq. 22–35. Additional assumptions implicit to the model structure are given in S4 Table.

### Formalizing secretory IgA (SIgA) action on microbial taxa

Our model capitalizes on SIgA's multifunctionality within the gut lumen, categorizing its actions based on its affinity level [48]. For consistency, we denote both the affinity of the antigen-binding site of a BCR on the B cells and the affinity of the SIgA produced by the plasma cells by $\rho_i$, where $i$ represents the taxon index for $1 \leq i \leq N$, and $N$ represents the number of taxonomic groups. SIgA may exert non-neutralizing but masking (M) actions, reflecting low-affinity antibodies, facilitating immune exclusion (preventing commensal bacteria from interacting with the immune system) and enhancing adherence to the gut lumen; and neutralizing (N) actions, reflecting high-affinity antibodies, contributing to the expulsion of bacteria from the gut lumen [17]. The activities of "masking" and "killing" are not mutually exclusive (Fig 2B), with their rates determined by the affinity levels:

$$\mu^m_{\ i} = \frac{1}{1 + \rho_i^{\overline{p}}},$$

(1)

$$\mu^n_{\ i} = 1 - \frac{1}{1 + \rho_i^{\overline{p}}},$$

(2)

where, $\mu^m_{\ i}$ and $\mu_i^{\ n}$ represent the masking and neutralizing rates of SIgA with affinity $\rho_i$, respectively. To differentiate the impact of maternal SIgA (mSIgA) from endogenous SIgA (eSIgA), we use four different parameters - $\mu_i^{m,\ n}$, $\mu_i^{m,\ m}$, $\mu_i^{e,\ n}$, and $\mu_i^{e,\ m}$—describing the neutralizing and masking rates of mSIgA and eSIgA for each taxon $i$.

Similar to their rate of action, the binding ability $\omega_i$ of the antibodies is determined by their respective affinity levels,

$$\omega_i = 1 - \frac{r_d}{1 + \rho_i},$$

(3)

where $r_d$ denotes the dissociation rate of SIgA, describing the rate at which the antibody dissociates from its target [82]. Masking and neutralizing rates are scaled by the binding ability for each taxon, which modulates their overall effectiveness. This dual function of SIgA creates three subpopulations for each microbial taxon $i$: uncoated ($y_i^{L,\ uc}$), masked ($y_i^{L,\ m}$), and neutralized, where the total population size that is metabolically active in the gut lumen is $y_i^L = y_i^{L,\ uc} + y_i^{L,\ m}$.

The exchange rate between $y_i^{L, m}$ and $y_i^{L, uc}$ depends on the level of mSIgA and eSIgA in the gut lumen, the half-life of SIgA ($l_{hl}$), the binding ability ($\omega_i$), and masking and killing rates ($\mu_i^{m, m}$, and $\mu_i^{e, m}$) (Eq. 26 and 27). Throughout the remainder of this text, SIgA−, SIgA+ (M), SIgA+ (N) will denote bacteria and/or antigens that are free of SIgA, forming complexes masked with SIgA, and neutralized by SIgA, respectively. Both masked and neutralized populations are coated with SIgA, meaning that techniques like IgA-Seq or BugFACS would detect SIgA binding in both cases. The SIgA coating ratio, which measures the proportion of bacteria coated with SIgA, would reflect contributions from both masked and neutralized bacteria.

## Characterizing the ecological interactions within the microbial community

The absolute abundance of each taxon ($y_i^L$) is governed by a generalized Lotka–Volterra (gLV) model [83], incorporating a net growth rate $\lambda_i$ and an interaction matrix $\beta$, where $\beta_{i,j}$ denotes the impact of the abundance of taxon $j$ on the abundance of taxon $i$. We assume that $y_i^{L, uc}$ and $y_i^{L, m}$ interact identically with their counterparts, therefore, the net change for each taxon is calculated based on the total abundance $y_i^L$ (Eq. 25), which is then divided into two subpopulations $y_i^{L, uc}$ and $y_i^{L, m}$ depending on the processes described above. While the gLV model simplifies higher-order dynamics by considering only pairwise interactions, this simplification is a well-established method that balances model complexity with the feasibility of parameter inference and has been successfully employed in several quantitative studies of gut microbiota dynamics [84–86].

In addition to gLV dynamics, factors in the gut lumen such as the oxygen concentration ($O_2$), caloric intake from HMOs, and caloric intake from PDPs directly modulate the growth rate of bacteria depending on their capabilities to proliferate in aerobic conditions ($\phi_i^{O2}$) and their efficiency in metabolizing different energy sources ($\phi_i^{HMOs}$, $\phi_i^{PDPs}$) as follows,

$$\lambda_i^{adj}(t) = \left(1 + O_2(t)\phi_i^{O_2} + HMOs(t)\phi_i^{HMOs} + PDPs(t)\phi_i^{PDPs}\right) \lambda_i,$$

(4)

where $\lambda_i^{adj}$ denotes the adjusted growth rate for taxon $i$. As a result of these dynamics, $O_2$—which is assumed to be present in the gut lumen at its maximum value after birth - is depleted at a constant rate ($\mu_{O2}$) by facultative anaerobes ($y_{fn}^L$) over time (Eq. 24). Caloric intake from HMOs and PDPs as well as the mSIgA levels depend on the EBF and MF durations, and the details of their parameterization will be discussed in the following sections.

## Modeling B-cell ontogeny

To characterize the dynamics underlying 'immune education', our model focuses on the ontogeny of the B-cell response of the infant, while implicitly integrating the role of the T-cell arm of humoral immunity. This requires formalizing an affinity maturation and plasma cell differentiation process that reflects BCR affinity on a continuous scale. To this end, we developed a set of differential equations that track the average affinity of the circulating B cells within the GCs and plasma cells in the Lamina Propria (LP). Our approach, inspired by previous work such as Molari *and colleagues* [87], adapts and extends their quantitative techniques to the specific context of gut mucosal immunity, incorporating continuous BCR affinity distributions and multiple thresholds for B cell fate decisions. Although previous stochastic models have explored GC dynamics [88–93], they primarily focus on vaccine development, systemic immunity, or model antibody affinity in a discretized manner, which limits their direct integration to our framework.

Circulating B cells originate from the naïve B cells migrating from bone marrow to the PPs, where they get activated upon antigen encounter. This activation rate depends on the linear combination of IgA-antigen immune complexes ($z_i^m$) and IgA-free antigens ($z_i^{uc}$) transported across M cells [33] on the PPs,

$$\psi_i = \psi^m z_i^m + \psi^{uc} z_i^{uc},$$

(5)

where the activation rates per sampled antigen for $z_i^m$ and $z_i^{uc}$ are denoted by $\psi^m$ and $\psi^{uc}$, respectively. $\psi_i$ represents the total activation rate dedicated to taxon $i$, and we assume that $\psi^{uc} > \psi^m$ due to the enhanced anti-inflammatory effects of $z_i^m$ compared to $z_i^{uc}$ [94,95] and thus the promotion of increased B cell recruitment and activation per $z_i^{uc}$ [96].

Five factors influence antigenic sampling by M-cells: (*i*) the developmental clock of the host, and associated functionality of GALT inductive sites, GCs, and M cells (which gradually increase over ontogeny) [14]; (ii) factors in the breastmilk that suppress the functionality of GALT inductive sites, GCs, and M cells [35]; (*iii*) microenvironmental stimulation ($\sigma$) representing the anti- or pro-inflammatory tone of the microenvironment, influencing barrier integrity [97] and M cell function [98]; (*iv*) the sampling rates of SIgA− and SIgA+ (M) bacterial antigens; and (v) invasiveness of each taxon ($\alpha_i$), which measures the bacteria's capacity to adhere to and penetrate intestinal epithelial cells [99]. After the host reaches the age at which their endogenous system becomes functional, $z_i^{uc}$ and $z_i^m$ can be expressed as,

$$z_i^{uc} = \sigma \, \alpha_i \in^{uc} y_i^{L,\, uc}, \tag{6}$$

$$z_i^m = \in^m y_i^{L,\, m}, \tag{7}$$

where $\in^{uc}$ and $\in^m$ denote the sampling rates of $y_i^{L,\, uc}$ and of $y_i^{L,\, m}$ for any taxon $i$, respectively. Given the M cells' binding capacity to SIgA [18,100,101], we assume that $\in^m > \in^{uc}$. Considering the role of SIgA binding in encouraging biofilm formation and thereby enhancing commensals' adherence to the mucosal surface in a beneficial manner [102–104], we assume that the sampling of $y_i^{L,\, uc}$ is influenced by both $\sigma$ and $\alpha_i$; whereas for $y_i^{L,\, m}$, sampling occurs more consistently, determined solely by the sampling rate $\in^m$.

## Role of the microenvironment

Microenvironment in the gut lumen and within the GALT inductive sites has a significant role in modulating the antigenic sampling and the affinity maturation process, respectively. Here, we define microenvironmental stimulation ($\sigma$) as a composite variable that encapsulates the cumulative effects of cytokines, microbial products, and host-derived signals on the local tissue milieu. The intestinal epithelial cells (IECs) that line the gut lumen possess a variety of receptors, including pattern recognition receptors (PRRs) like Toll-like receptors and NOD-like receptors, which can detect microbial-associated molecular patterns from commensal or pathogenic microbes [105,106]. Upon sensing these signals, IECs produce and release a range of signaling molecules, including cytokines and chemokines that modulate immune responses [107]. Considering this signal transmission pathway, we define microenvironmental stimulation as,

$$\sigma = \sum_i \kappa_i y_i^{L,\, uc} - \sum_i l_i y_i^L, \tag{8}$$

where $\kappa_i$ represents the relative immunostimulatory capacity of the SIgA- members of taxon $i$ compared to their SIgA+ (C) counterparts. $l_i$ is a binary variable representing the inherent anti-inflammatory potential of taxon $i$, including mechanisms such as short-chain fatty acid production, modulation of immune cell activity, or enhancement of epithelial barrier function [108–111]. Assigning a binary value to $l_i$ and continuous value to $\kappa_i$ allows us to focus solely on quantifying the *relative* immunostimulatory capacity of unmasked commensals [40,41], simplifying the parameter inference procedure discussed in the subsequent sections. Our framing captures how commensals can either reduce inflammation or, conversely, display pathogenic-like behavior when gut homeostasis is disrupted and the regulatory effect of SIgA masking on bacterial populations is compromised. For commensals without any anti-inflammatory capacity, $l_i$ would be zero, meaning that their masked counterparts are irrelevant in modulating the microenvironment. Thus, in the extremes, exclusively probiotic bacteria would be assigned $l_i = 1$ and $\kappa_i = 1$, meaning that their unmasked counterparts do not cause any increase in the

inflammatory tone, whereas their masked counterparts decrease inflammation proportional to their abundance in the gut lumen ($\sigma = \sum_i \kappa_i y_i^{L,\ uc} - \sum_i l_i y_i^L = \sum_i (\kappa_i - l_i) y_i^{L,\ uc} - \sum_i l_i y_i^{L,m} = -\sum_i y_i^{L,m} \mid (\kappa_i = l_i) = 1$). In contrast, exclusively pathogenic bacteria would have $l_i = 0$ and a high $\kappa_i$ underscoring their substantial capacity to induce inflammation. To reduce the impact of identifiability problems during inference, we fixed $\kappa_i = 1$ for all the symbiotic commensals (*Bifidobacteriaceae*, *Bacteroidaceae*, and *Clostridiales*) in our model, and only inferred $\kappa_E$ for *Enterobacteriaceae*.

Optimal regulation of $\sigma$ is essential for maintaining gut homeostasis; however, when excessive, it can compromise the integrity of the intestinal barrier by disrupting tight junctions between epithelial cells, leading to altered M cell function either by direct damage or through influencing the M cell differentiation and function [98]. Therefore, $\sigma$ is used as an effect modulator in Eq. 6.

Elevated levels of stimulation in the lumen can trigger IECs to produce pro-inflammatory signals directed toward essential immune cells involved in B cell maturation, including DCs and T cells located in the subepithelial dome of the GALT inductive sites. While recognizing that the lumen and the distinct zones within the GALT inductive sites exhibit specific microenvironmental characteristics, we assume that the microenvironmental tone in the lumen is mediated through the epithelium [42,112] and influences the tone in the GALT inductive sites in a similar fashion. Thus, we employ $\sigma$ to simultaneously describe the microenvironmental stimulation in the gut lumen and within the GALT inductive sites.

## Modeling the role of germinal centers

The microenvironment within the GALT inductive sites plays a crucial role in steering the balance between T follicular helper (Tfh) and follicular regulatory (Tfr) cells [113,114]. This balance is particularly important to select the pool of circulating B cells with the appropriate BCR affinity distribution to continue through each P-SHM-S cycle. However, this conventional view of affinity maturation does not allow for antigen-specific low-affinity clones to survive the competition for T cell help in the light zone. An alternative view suggests that the continually available antigen and sufficient T-cell help (possibly of a range of types) cooperating with commensal-derived signals acting via PRRs supports the survival and differentiation of low-affinity cells as well as the high-affinity ones [115]. Therefore, assuming that the circulating B cells constitute a normally distributed set of BCR affinities [87], we propose a variable reflecting the Tfh:Tfr ratio which determines the BCR affinity *range* that will receive adequate T cell help to continue with the next SHM cycle, and eventually the average affinity of the plasma cells leaving the GC. We refer to this quantity as the selection threshold, denoted by $\delta_i$.

In addition to the microenvironment, antigen presentation is pivotal in determining the Tfr versus Tfh differentiation via conditioning of mucosal DCs [116,117]. SIgA-coated bacterial antigens promote the tolerogenic programming of DCs via the C-type lectin receptor SIGNR1 [75]. Given that the sampling of SIgA-antigen immune complexes will skew the bias in differentiation toward tolerogenic profiles compared to SIgA-free antigens [94,118], we model the rate of change in $\delta_i$ as a function of the ratio of cumulative uncoated ($z_i^{\sum uc}(t) = \sum_t z_i^{uc}(t)$) to cumulative total sampled antigens ($z_i^{\sum}(t) = \sum_t (z_i^{uc}(t) + z_i^c(t))$), as well as the microenvironmental stimulation $\sigma$ (Eq. 28), as a proxy to reflect cumulative imprinting of the Tfh:Tfr ratio. This formulation allows us to implicitly model the impact of T cells on GC reactions, although we do not explicitly include them in the differential equation system.

We model one cycle of GC reactions as follows,

1. At each timestep $t$, a population of naïve B cells ($B_i^n$) in the number of $\Delta B_i^n$ migrate from bone marrow to the dark zone PPs. As for T cells, the rate of migration is modeled as an exponentially decreasing function of time $t$ ($C_n e^{-c_n t}$), reflecting the diminishing pool of naïve cells as the host ages (Eq. 29) [14,119]. Although naïve cells are not antigen-specific, for the sake of completeness, we use the taxon subscript $i$ to track the number of B cells activated in an antigen-specific fashion. These cells lay dormant in the GALT inductive sites until the M cells and GCs mature at time $t^m$.

2. At time $t^m$, $B_i^n$ starts getting activated at the rate of $\psi_i$ (Eq. 5) and join the pool of circulating B cells ($B_i^c$) participating in the GC reactions. These newly activated cells are calculated as $B_i^{c,\ new} = \psi_i B_i^n$.

3. Newly activated population of circulating B cells have a normal BCR affinity distribution with mean 0 and a standard deviation proportional to their population size, providing enough BCR variability for the selection process in the light zone.

4. Given the value of $\delta_i$, selection pressure imposed by T cells segregates the circulating cells in the light zone into 4 different groups,

   4.1. Cells with BCR affinity $<th_{apop}\delta_i$ fail to compete for T cell help and die through apoptosis. The number of these cells is equal to $B_i^c\Phi((th_{apop}\delta_i - \bar{\rho_i}^c)/\sigma_i^c)$, where $\Phi$, $\bar{\rho_i}^c$, and $\sigma_i^c$ denote the cumulative distribution function for normal distribution, the mean, and the standard deviation of the $B_i^c$ BCR affinity distribution, respectively.

   4.2. Cells with BCR affinity in the range of $[th_{apop}\delta_i, th_{high}\delta_i]$ will receive enough T cell help to continue circulating and will move to the dark zone for the next round of proliferation and SHM cycle in the dark zone. The number of these cells is equal to $B_i^c(\Phi((th_{high}\delta_i - \bar{\rho_i}^c)/\sigma_i^c) - \Phi((th_{apop}\delta_i - \bar{\rho_i}^c)/\sigma_i^c))$.

   4.3. Cells with BCR in the range of $[th_{high}\delta_i, th_{ang}\delta_i]$ will receive enough T cell help to terminally differentiate to plasma cells with the average affinity of $0.5(th_{ang} + th_{high})\delta_i$. The number of these cells is equal to $B_i^c(\Phi((th_{ang}\delta_i - \bar{\rho_i}^c)/\sigma_i^c) - \Phi((th_{high}\delta_i - \bar{\rho_i}^c)/\sigma_i^c))$.

   4.4. Cells with BCR affinity above $th_{ang}\delta_i$ will be rendered anergic through additional regulatory mechanisms, mirroring the principles of negative selection that eliminate self-reactive B cells [120]. The number of these cells is equal to $1 - B_i^c\Phi((th_{ang}\delta_i - \bar{\rho_i}^c)/\sigma_i^c)$.

   This selection process confines the BCR affinity range of circulating cells to $[th_{apop}\delta_i, th_{high}\delta_i]$ prior to the next SHM cycle. Although the process involves truncating a Normal distribution, for the sake of analytical computability, we assume that the resulting population maintains a normal distribution with mean $0.5(th_{apop}\delta_i + th_{high}\delta_i)$ and standard deviation $(1/6)(th_{high}\delta_i - th_{apop}\delta_i)$, indicating that 99.7% of the affinity levels lie within the $[th_{apop}\delta_i, th_{high}\delta_i]$ range.

5. For $t > t^m$, circulating cells that received enough T cell help at the previous time step $(t - \Delta t)$ will migrate back to the dark zone, proliferate with rate $r_p$, and go through another round of SHM. We assume that the proliferation phase only changes the size of the $B_i^c$ pool, whereas SHM increases variability in the circulating BCR affinity pool. Each round of SHM increases the standard deviation of the circulating BCR affinity distribution proportional to $\tau^c|\delta_i(t+1) - \bar{\rho_i}^c(t)|$ to ensure that there is enough variation in the circulating BCR affinities, allowing the model to mimic the ranking-based selection imposed by the T cells in the light zone of the GCs [88].

6. Proliferated and mutated cells will be pooled together with the newly activated cells $B_i^{c,\ new}$, which will modify the pool size as well as the mean and the standard deviation of the BCR affinity distribution. Since our model is based on differential equations, we keep track of the difference of the circulating pool size, the mean, and the standard deviation of the circulating BCR affinity pool between consecutive timesteps. We approximate the pooled mean and standard deviation as a weighted sum by the sample sizes of the already circulating and newly added cells, and calculate the differences as,

$$\Delta B_i^c(t + \Delta t) = B_i^c(t)\left[\Phi\left(\frac{th_{high}\delta_i(t) - \bar{\rho_i}^c(t)}{\sigma_i^c(t)}\right) - \Phi\left(\frac{th_{apop}\delta_i(t) - \bar{\rho_i}^c(t)}{\sigma_i^c(t)}\right)\right](1 + r_p) + B_i^{c,\ new}(t) - B_i^c(t),$$

(9)

$$\Delta\bar{\rho_i}^c(t)(t + \Delta t) = \frac{B_i^c(t)\left[\Phi\left(\frac{th_{high}\delta_i(t) - \bar{\rho_i}^c(t)}{\sigma_i^c(t)}\right) - \Phi\left(\frac{th_{apop}\delta_i(t) - \bar{\rho_i}^c(t)}{\sigma_i^c(t)}\right)\right]\int_{th_{apop}}^{th_{high}} N(\bar{\rho_i}^c(t),\ \sigma_i^c(t))\ \partial\rho}{B_i^c(t)\left[\Phi\left(\frac{th_{high}\delta_i(t) - \bar{\rho_i}^c(t)}{\sigma_i^c(t)}\right) - \Phi\left(\frac{th_{apop}\delta_i(t) - \bar{\rho_i}^c(t)}{\sigma_i^c(t)}\right)\right](1 + r_p) + B_i^{c,\ new}(t)} - \bar{\rho_i}^c(t),$$

(10)

$$\Delta\sigma_i^c(t+\Delta t) = \frac{B_i^c(t)\left[\Phi\left(\frac{th_{high}\delta_i(t)-\bar{\rho}_i^c(t)}{\sigma_i^c(t)}\right) - \Phi\left(\frac{th_{apop}\delta_i(t)-\bar{\rho}_i^c(t)}{\sigma_i^c(t)}\right)\right](1+r_p)\left(\frac{1}{6}\right)[th_{high}\delta_i - th_{apop}\delta_i] + B_i^{c,\ new}(t)\sigma_i^{new}}{B_i^c(t)\left[\Phi\left(\frac{th_{high}\delta_i(t)-\bar{\rho}_i^c(t)}{\sigma_i^c(t)}\right) - \Phi\left(\frac{th_{apop}\delta_i(t)-\bar{\rho}_i^c(t)}{\sigma_i^c(t)}\right)\right](1+r_p) + B_i^{c,\ new}(t)} - \sigma_i^c(t),$$

(11)

where $\mathbb{N}(x\,|\,\mu,\ \sigma)$ denotes the Normal probability distribution function with mean $\mu$ and standard deviation $\sigma$, and $\sigma_i^{new} = \tau^{new}B_i^{c,\ new}$ as discussed in step 3 above.

7. The same rationale applies when calculating the difference of the size of the plasma cell pool ($B_i^p$) and the mean of the plasma BCR affinity pool ($\bar{\rho}_i^p$) between consecutive timesteps

$$\Delta B_i^p(t+\Delta t) = B_i^p(t)\left[\Phi\left(\frac{th_{ang}\delta_i(t)-\bar{\rho}_i^c(t)}{\sigma_i^c(t)}\right) - \Phi\left(\frac{th_{high}\delta_i(t)-\bar{\rho}_i^c(t)}{\sigma_i^c(t)}\right)\right](1-B_i^p(t)),$$

(12)

$$\Delta\bar{\rho}_i^p(t+\Delta t) = \frac{B_i^p(t)\left[\Phi\left(\frac{th_{ang}\delta_i(t)-\bar{\rho}_i^c(t)}{\sigma_i^c(t)}\right) - \Phi\left(\frac{th_{high}\delta_i(t)-\bar{\rho}_i^c(t)}{\sigma_i^c(t)}\right)\right](0.5\,(th_{ang}+th_{high})\,\delta_i(t)) + B_i^p(t)\bar{\rho}_i^p(t)}{B_i^p(t\left[\Phi\left(\frac{th_{ang}\delta_i(t)-\bar{\rho}_i^c(t)}{\sigma_i^c(t)}\right) - \Phi\left(\frac{th_{high}\delta_i(t)-\bar{\rho}_i^c(t)}{\sigma_i^c(t)}\right)\right]B_i^p(t)} - \bar{\rho}_i^p(t).$$

(13)

Note that $\Delta B_i^p(t+\Delta t)$ is multiplied by $(1-B_i^p(t))$ to ensure that the plasma cell pool saturates at the carrying capacity of 1, reflecting the carrying capacity for plasma cells in the LP [121].

We assume that the longevity of plasma cells is primarily determined by the cumulative T cell help they have received at the time they left the GC reaction [122]. Since the affinity levels (or magnitude of $\delta_i$) do not necessarily equate to the cumulative T cell help (or the number of SHM-S-P cycles) itself, we employ delayed differential equations to track the relative progression of $\delta_i(t)$, and define the death rate of plasma cells as $\mu_i^p = \frac{\delta_i(t-1)}{\delta_i(t)}$. By doing so, we introduce a higher death rate to the plasma cells that were produced at the early stages of immune ontogeny, and as the host matures and $\delta_i(t)$ approaches $\lim_{t\to\infty}\delta_i(t)$, $\mu_i^p$ approaches 0.

In reality, plasma cells do not live infinitely long; therefore, a $\mu_i^p$ of 0 would not be correct. However, plasma cells are continuously replaced by the memory cells, which we do not include in our model (S1 Text, see section *Model Limitations*). Therefore, $\mu_i^p = 0$ at the Steady Phase mimics the continuous replacement of plasma cells by the memory cells, and thus the plasma cell population is kept at a steady level.

### Linking model outputs to data on microbial taxa

Our model quantifies the absolute abundances of microbial populations within the gut lumen. However, for parameter inference, we relied on fecal samples from infants, analyzed using 16S rRNA sequencing techniques [20–22]. Fecal samples serve as a proxy for assessing the taxonomic distribution within the gut lumen. However, equating these two environments directly overlooks the distinct roles of SIgA—specifically, its ability to either neutralize bacteria by facilitating their expulsion from the gut lumen [43] or, conversely, enhancing their adherence through masking without neutralization [46,47]. To incorporate this dual functionality, we introduce 3 different observation rates $s^n$, $s^m$, and $s^{uc}$ for neutralized, masked, and uncoated subpopulations in the fecal samples, assuming that the observed quantity is the sum of these subpopulations weighted by their respective observation rates. We keep track of the cumulative neutralized bacteria abundance ($dy_i^{L,\ \sum n}$) to quantify the total bacterial population in the fecal samples, although they are assumed to remain inactive, neither interacting with other bacteria nor stimulating the immune system. During parameter inference, we reconstruct the absolute abundances in fecal samples as follows,

$$w_i^n = s^n\frac{dy_i^{L,\ \sum n}}{dt},$$

(14)

$$w^m{}_i = s^m y_i^{L,\,m},$$ (15)

$$w^{uc}{}_i = s^{uc} y_i^{L,\,uc},$$ (16)

$$w_i = w^n{}_i + w^m{}_i + w^{uc}{}_i,$$ (17)

where $w^n{}_i$, $w^m{}_i$, $w^{uc}{}_i$, and $w_i$ denote the neutralized, masked, uncoated, and total absolute abundances observed per fecal content for taxon $i$, respectively.

## Informing time-varying components of the model

Our model mainly focuses on the mechanisms governing the GC reactions in the GALT inductive sites, and the main route of antigen transport to these sites is carried out by M cells which appear shortly before weaning primarily due to the decreasing concentrations of maternal steroids in the breastmilk [33,35]. In the absence of quantitative data, we propose using the concentration of mSIgA as an indirect measure of maternal steroid levels. We assume that antigenic sampling beings as $t^{50}$, representing the time point when mSIgA concentration falls below 50% of its peak value, favoring the qualitative observation regarding the close timing of M cell maturation to weaning (Fig 1A). In the absence of breast-feeding, various factors—including the maturation of DCs and BCRs—may constrain the endogenous immune response, notably affecting the development of IgA-secreting plasma cells [123–125]. Observations in the literature suggest that these constraints are particularly pronounced during the first month of human life, and infants who do not receive human milk compensate by starting to produce SIgA in the intestine around 4 weeks of age [44]. Therefore, we assume that the endogenous immune system activation occurs at day $t^m = (t^{50}, 30)$, allowing us to compute $t^m$ for a generic combination of EBF and MF durations.

To simulate our system for any given feeding pattern, we parameterize mSIgA, HMOs, and PDPs as functions of EBF and MF durations denoted by $T_{EBF}$ and $T_{MF}$, respectively. To do so, we used the estimated calorie requirements of infants [25–27], and assumed that (i) average calories in breastmilk per volume is constant through the period of breastfeeding and the caloric needs are met by modulating the volume ingested by the infant, (ii) total calorie requirement of the infant for a given time point is always met regardless of the feeding pattern, and (iii) when the infant starts MF, HMOs are reduced and PDPs are introduced gradually (Fig 2A). Upon switching to MF at time point $t_{MF}$, this pattern resumes a gradual increase with a notable rise in the proportion of calories derived from PDPs. We formulate the function for HMOs caloric intake as,

$$HMOs(t) \overset{\text{def}}{=} f_{HMOs}(t, t_{MF}, T_{MF}) = \{K_{H1} + V_0\left(1 - e^{-c_H t}\right),\; K_{H2}(1 - e^{-c_H t_{mirror}}),\; t < t_{MF}\; t \geq t_{MF},$$ (18)

where $t_{mirror} = \dfrac{t_{MF}[T_{MF} - (t - t_{MF})]}{T_{MF}}$ represents the adjusted time points between $t_{MF}$ and the switch to ECF. $K_{H2} = \dfrac{K_{H1} + V_0(1 - e^{-c_H t_{MF}})}{1 - e^{-c_H t_{MF}}}$, ensuring a smooth transition at $t = t_{MF}$. To calculate from PDPs, we first formulate the function for the total caloric intake $f_{TOT}$ as,

$$f_{TOT}(t, t^*{}_{MF}, T^*{}_{MF}) = \left\{ f_{HMOs}(t, t^*{}_{MF}, T^*{}_{MF}),\; \frac{K_C}{1 + e^{-c_c(t - t^*{}_{MF})}},\; \frac{K_C}{1 + e^{-c_c(t_C - t^*{}_{MF})}},\; t < t^*{}_{MF}\; t^*{}_{MF} \leq t < t_C\; t \geq t_C, \right.$$ (19)

where $t^*{}_{MF}$ and $T^*{}_{MF}$ denote the $t_{MF}$ and $T_{MF}$ values derived from the study that provided data for our model parameterization [18]. $K_c$ denotes the fixed caloric intake of the host when they reach the steady phase at $t = t_c$, and $c_c$ adjusts the rate of increase that intake until $t = t_c$. Eq. 19 ensures that the total caloric intake of the infant for any given $\{t_{MF}, T_{MF}\}$

combination matches the total caloric intake in the study used for model parameterization. Given Eq. 18 and 19, we can compute the caloric intake from PDPs as,

$$PDPs(t) \overset{\text{def}}{=} f_{PDPs}(t, t_{MF}, T_{MF}) = \{0, \ f_{TOT}(t, t^*_{MF}, T^*_{MF}) - f_{HMOs}(t, t_{MF}, T_{MF}), \ t < t_{MF} \ t \geq t_{MF},$$

(20)

Data in literature indicate that SIgA levels in breastmilk peak in the days immediately postpartum, followed by a decline over the first month, eventually stabilizing into a lower, steady state in mature milk [113–115]. Consequently, we formulate and parameterize the rate of change in mSIgA levels in the breast milk as,

$$\Delta f_{mSIgA}(t) = K_l + m_l e^{-c_l t},$$

(21)

where $K_l$ represents the steady-state influx rate of mSIgA into the gut lumen as found in mature milk. Parameters $m_l$ and $c_l$ modulate the mSIgA level shortly after birth and its subsequent decline. This rate is modulated by $HMOs(t)$ as it is used as a proxy for the volume of breastmilk ingested by the infant (Eq. 22).

### Differential equation system

#### Microenvironmental dynamics.

$$\frac{dmSIgA}{dt} = +HMOs(t)\Delta f_{mSIgA}(t) - I_{hl}mSIgA.$$

(22)

$$\frac{deSIgA}{dt} = +C_l B_i^p \left(1 - \left(1 - \frac{I_{hl}}{C_l}\right)eSIgA\right) - I_{hl}eSIgA.$$

(23)

$$\frac{dO_2}{dt} = -\mu_{O2}O_2 \ y_{fn}{}^L.$$

(24)

#### Community dynamics.

$$\frac{dy_i^L}{dt} = y_i^L(\lambda_i^{adj} + \beta_i) - (\omega_i^{m,\ n}\mu_i^{m,\ n} + \ \omega_i^{e,\ n}\mu_i^{e,\ n})y_i^{L,\ uc}, \text{where } \beta_i = \ \sum_{j=1}^N \beta_{i,j} \ y_j^L$$

(25)

$$\frac{dy_i^{L,\ uc}}{dt} = \frac{dy_i^L}{dt} + (1/I_{hl})y_i^{L,\ c} - (\omega_i^{m,\ c}\mu_i^{m,\ c} + \ \omega_i^{e,\ c}\mu_i^{e,\ c})y_i^{L,\ uc}.$$

(26)

$$\frac{dy_i^{L,\ c}}{dt} = -(1/I_{hl})y_i^{L,\ c} + (\omega_i^{m,\ c}\mu_i^{m,\ c} + \ \omega_i^{e,\ c}\mu_i^{e,\ c})y_i^{L,\ uc}.$$

(27)

$$\frac{dy_i^{L,\ \sum n}}{dt} = +(\omega_i^{m,\ n}\mu_i^{m,\ n} + \ \omega_i^{e,\ n}\mu_i^{e,\ n})y_i^{L,\ uc}.$$

(28)

**Immune dynamics.** To make our model neutral to the contribution of $\sigma$ versus $\frac{z_i^{\sum uc}(t)}{z_i^{\sum}(t)}$ in Eq. 29 below, we introduce $\tau^\sigma$ as a multiplicative factor such that $max_t \tau^\sigma \sigma(t) = 1$. Due to the dynamic nature of our model, analytically calculating $max_t \ \sigma(t)$ is not possible. However, the upper bound of this quantity can be calculated by using Eq. 8 and 26 by assuming all bacteria in the lumen are uncoated and have reached their carrying capacity in the absence of any neutralization by antibodies. However, this calculation would require a numerical integration for a multi-taxa system. We can further

approximate the upper bound by only using the steady state value of the pathogenic taxon in our model, which has the highest $\kappa_i$ that is orders of magnitude larger than the $\kappa_i$ for symbiotic commensals, and approximate $\tau^\sigma = 1/(\kappa_i \lambda_i / \beta_{ii})$.

$$\frac{d\delta_i}{dt} = +\tau^\delta\, C_T e^{-c_T t}\, \log\left(1 + \tau^\sigma \sigma\right) \log\left(1 + \frac{z_i^{\sum uc}(t)}{z_i^{\sum}(t)}\right). \tag{29}$$

$$\frac{dB_i^n}{dt} = +\, C_B e^{-c_B t} - \psi_i B_i^n. \tag{30}$$

$$\frac{dB_i^c}{dt} = +\Delta B_i^c(t + \Delta t). \tag{31}$$

$$\frac{dB_i^p}{dt} = +\Delta B_i^p(t + \Delta t) - \mu^p{}_i B_i^p. \tag{32}$$

$$\frac{d\bar{\rho}_i^c}{dt} = +\Delta\,\bar{\rho}_i^c(t + \Delta t). \tag{33}$$

$$\frac{d\bar{\rho}_i^p}{dt} = +\Delta\,\bar{\rho}_i^p(t + \Delta t). \tag{34}$$

$$\frac{d\sigma_i^c}{dt} = +\Delta\sigma_i^c(t + \Delta t). \tag{35}$$

## Parameter inference

Given the complexity of our model, we incorporated as many publicly available datasets as possible to inform the inference of all model parameters and to minimize potential identifiability issues. This included integrating diverse datasets, including longitudinal data on relative abundances (ratios) [20], overall density of bacteria (cells/g of stool) [23], cross-sectional data on SIgA indexes [21], SIgA coating ratio [36], and the varying abundance of taxa with different feeding practices [22]. We have used the published abundance results on a taxonomic level. While we acknowledge that these data sources differ in sample origin, sequencing regions, and database references, they still represent the most comprehensive data available for our analysis. S1 Table summarizes the use of these data sources alongside their methods of taxonomic classification. Data used for model fitting can be found at https://doi.org/10.5281/zenodo.15629746.

The complexity of our model relative to the volume of data posed challenges in simultaneously inferring all parameters. Therefore, we initially focused on the Maternal and Steady phases—time periods when the host's endogenous system is either not active or fully matured, respectively. This approach allowed us to ignore the parameters related to the developmental phase at this stage of inference and focus on a smaller set of shared parameters between the Maternal and Steady phases. Using the parameters estimated from this initial stage, we then proceeded to infer parameters relevant to the developmental phase.

**Joint parameter inference of the maternal and the steady phase.** The key observation which made this inference stage possible is the multigenerational transmission of the mucosal immune response [21], that the offspring converges to a similar SIgA response as the mother, i.e., $\mu_i^{e,\,n}(t) = \mu_i^{m,\,n}(t)$ and $\mu_i^{e,\,m}(t) = \mu_i^{m,\,m}(t)$ for $t \geq 720$. Accordingly, we only need to infer one set of SIgA affinity values that are shared between the mother and the matured offspring. Since this

stage of inference does not include the developmental phase, we cannot quantify the eSIgA levels as they are dynamically calculated via Eq. 23, and thus assume that the eSIgA concentration reaches its carrying capacity of 1 for the matured offspring.

We employed Bayesian statistical analysis using the RStan software (Version 2.32.3; Stan Development Team) [126], and computed a composite likelihood function based on the simultaneous optimization of four different signals: (*i*) relative abundances between days 3 and 274 representing the maternal phase (given the metadata, the Maternal phase ends at day 172 when averaged over all subjects. After the data is interpolated and smoothed for inference, this time period was cut to 154 days. We then account for an additional 120 days adding up to 154 + 120 = 274 days for the symbiotic commensals (*Bifidobacteriaceae*, *Bacteroidaceae*, *Clostridiales*) to capture their interaction terms. However, we strictly used 172 days for the pathogenic taxon *Enterobacteriaceae* to avoid the impact of interaction to be mistaken for the neutralizing effects of the endogenous immune system starting to develop after day 154), (*ii*) relative abundances between days 721 and 735 representing the steady phase,(*iii*) relative abundances at day 30 in case of no breastfeeding, and (*iv*) SIgA indexes at the steady phase (to align with the feeding practices of the cohort *Tsuduka and colleagues*, we filtered the subjects in this dataset using the metadata provided on the ratio of breastmilk to formula feeding (B/F). We used a lower bound of 0.75 and an upper bound of 0.25 for the average B/F over the Maternal and the Developmental Phase, respectively, and picked the subjects who satisfied both criteria. This led to 154 days of EBF and 308 days of MF on average). Error is assumed to be normally distributed for all signals. To manage signals of varying lengths and prevent overfitting to any particular signal, we established a precomputed upper bound for the likelihood of each signal, taking into account their length and the standard deviation applied in their likelihood calculations. Subsequently, we normalized the likelihood of each signal against this upper bound, capping it at 1 if it surpassed this limit, to ensure no single signal is disproportionately optimized. Parameters relevant to this inference stage are denoted by "Inferred-1" in S3 Table with their respective prior distributions.

**Parameter inference of the developmental phase.** Using the parameters inferred from the previous stage, we move onto inferring parameters relevant for the developmental phase, denoted by "Inferred-2" in S3 Table with their respective prior distributions. We computed a composite likelihood function based on the SIgA affinities ($\rho^{\bar{p}}_i(t)$) and the total SIgA coating ratio of the microbiome observed in the fecal samples of the mature host [36], i.e., $\frac{w^m_i(t) + w^n_i(t)}{w_i(t)}$ for $t \geq 720$. Due to the non-smooth nature of the functions involved in this phase, we employ a grid-search algorithm [127] to infer the parameters to optimize the likelihood function.

**Data curation and pre-processing.** We used two datasets based on relative abundance data derived from stool samples during inference. First dataset is the 16S rRNA sequencing analysis combined with feeding metadata presented in *Tsukuda and colleagues* [20], which is based on longitudinally collected fecal samples from 12 subjects during the first 2 years of life. Based on their functional significance in determining the community profiles during early life, we have grouped our focal genera into four distinct taxonomic groups: *Enterobacteriaceae* (*E*), *Bifidobacteriaceae* (*B*), *Bacteroidaceae* (*BC*), and *Clostridiales* (*C*) (S2 Table). Each group was selected for its unique role in influencing the model dynamics: *E* encompasses potentially pathogenic bacteria (genera *Escherichia* and *Shigella*) encountered pre-weaning when the endogenous system is more vulnerable to enteric infections. *B* represents the early colonizers with anti-inflammatory properties, regulating the microenvironment and providing colonization resistance. *BC* and *C* (*Anaerostipes*, *Blautia*, *[Eubacterium] hallii* group, *Faecalibacterium*, and *Ruminococcus*, all of which are non-pathogenic commensal bacteria contributing significantly to gut homeostasis) are the post-weaning bacteria with potential anti-inflammatory properties, which, unless regulated by SIgA, may stimulate the gut environment. We recognize that these taxonomic groups are not homogenous, and while some variability exists within each group—particularly under dysbiotic conditions—the classification reflects general trends observed in microbial behavior and immune modulation. Thus, although individual strains may diverge in their effects on gut homeostasis, this grouping provides a meaningful basis for modeling microbial-immune interactions.

Relative abundances provided in [20] are interpolated and smoothed using the loess function in R with a span parameter of 0.5, and weighed with a sigmoid function centered around the day of transition to MF (day 172 when averaged over all subjects based on the metadata) with a scale parameter of 5 to ensure a smooth transition from maternal to developmental phase (details can be found in https://doi.org/10.5281/zenodo.15629746, SMOOTH_DATA.R). These smoothed relative abundances are then converted to absolute abundances using data regarding fecal density counts (cells/g fecal content) [23], assuming that the total bacterial load stabilizes after 2 years (data extracted from Fig 2 in [23]). Although $BC$ and $C$ were detected at low abundances in pre-weaning samples, we assume that their introduction primarily occurs with the transition to MF, thus, their abundances are considered negligible and set to zero in the maternal phase. We assume that the net growth rate encompasses the net influx of bacteria, and introduce the taxa once at their respective time point as modifications to the initial conditions of the differential equation system. Consequently, depending on the duration of EBF, MF, and the values of $HMOs(t)$ and $PDPs(t)$, we adjust the initial conditions for $B$ and $E$ at the start of the Maternal, and for $BC$ and $C$ at the start of the Developmental Phase. We assume that abundance of $B$ transferred from mother to offspring exponentially decreases to 25% of its maximum value as the EBF duration shortens from 3 to 0 days, mimicking the high bacterial counts in colostrum and its impact on initial microbial seeding [128] (to adjust the abundance of *Bifidobacteriaceae* for a given duration of EBF including the durations shorter than 3 days, we multiply the abundance that is transferred for the feeding patterns provided in the metadata with a multiplier of $0.25 + 0.75(1 - \exp(-2.207t))$, ensuring a decreasing increase from 0.25 to 1 between days 0 and 3 and convergence to 1 for day 3 and onwards). $BC$ and $C$ abundances at the start of the Developmental Phase are determined by the first non-zero value they take at time $t = t^{*}_{MF}$, and are scaled by $PDPs(t_{MF})/PDPs(t^{*}_{MF})$ to adjust the inoculum size introduced with MF accounting for the deviations from the dataset used for inference.

The second dataset by Pan *and colleagues* [22] consists of 16S rRNA sequencing data from infants categorized by feeding type: breastfed, partially breastfed, and formula-fed. Due to the lack of detailed information on the partially breastfed group, we focus on the cross-sectional abundance data from formula-fed infants to impose additional constraints on the inference procedure. Relative abundances of $E$ (~32%), $B$ (~21%), $BC$ (~23%), and $C$ (~12%) provided in [22] for formula fed infants at day 30 of life (https://doi.org/10.5281/zenodo.15629746, under/PAN_DATA/Figure2a_quantified.png) are scaled according to the total relative abundance of $E + B + BC + C$ (72%) at day 29 provided in [20] and incorporated into the composite likelihood function representing the scenario of no breastfeeding. Including this scenario provides valuable information to the inference process as to how different calorie sources (HMOs and PDPs) impact the growth rate of different taxa.

IgA indexes of human twins provided in [21] are filtered according to the 'Ratio of Breast milk: Formula' column in the dataset. For the EBF period (until day 154, time of weaning in our model), 'Ratio of Breast milk: Formula' >0.75 is used. For the MF period (after day 154) this constraint is changed to 'Ratio of Breast milk: Formula' <0.5 to ensure that the subjects' diet timeline aligns with the ones in [20]. The code for processing the data provided in [21] is available in our repository (https://doi.org/10.5281/zenodo.15629746, DATA_PLANER.R).

Differential growth capabilities and $O_2$ metabolism is provided in S2 Table [28–31,57]. We assumed a negative directionality for all intra-taxa competition terms and for the inter-taxa competition terms between the set $\{B, BC, C\}$ and $E$ [84] and informed the estimation of $\beta_{i,j}$ accordingly. However, determining the directionality of the remaining inter-taxa terms is not straightforward due to multiple mechanisms simultaneously dictating these interactions, such as competition and cross-feeding [129–133], thus, they were not informed *a priori*.

**Inference results and values of model parameters.** We use three different quantification types to indicate how parameters are quantified: (*i*) "Inferred-1" indicates that the value of the parameter is estimated during the joint parameter inference of the Maternal and the Steady Phase, (i*i*) "Inferred-2" indicates that the value of the parameter is estimated during the joint parameter inference of the Developmental Phase, (*iii*) "Dependent" indicates that the quantification of the parameter depends on the value of other parameters, (*iv*) "Assumed Based on Literature" (ABL) refers to parameters for

which quantitative values are derived from other quantitative or qualitative data sources and expert opinion available in the literature, and (*v*) "Calibrated" refers to parameters that are tuned to achieve the values quantified via the former first four quantification methods. Similar quantification methods apply to parameterize the prior distributions used during inference, where they are either ABL or calibrated.

Given the absence of quantitative data for specific variables and parameters within our system, we employ normalization to assess their temporal influence on model dynamics. This approach applies to variables, such as $O_2$, $mSIgA$, and $eSIgA$. Their impact on the system is maximized when their normalized value reaches 1 and minimized at the value of 0.

## Supporting information

**S1 Fig. Inference results of RStan.** Two chains are used with 500 and 1,000 for warm-up and total iterations, respectively. Twenty-four of 2000 (2.4%) transitions ended with a divergence. (**A**) Relative abundance estimate results for the maternal phase, for 154 days of exclusive breastfeeding (EBF) and 308 days of mixed feeding (MF). (**B**) Relative abundance estimate results for the maternal phase, for 308 days of MF with no EBF. (**C**) Relative abundance estimate results for the steady phase, for 154 days of EBF (EBF) and 308 days of MF. Shaded areas represent the 95% confidence intervals. The data underlying this figure can be found in https://doi.org/10.5281/zenodo.15629746.
(TIF)

**S2 Fig. Heatmap of normalized predictor importance values across time points from random forest models.** Each cell shows the normalized importance of a predictor in explaining total fecal abundance of *Enterobacteriaceae* at a given time point (in days), as determined by conditional permutation importance in a random-forest model. Predictors include eSIgA affinity and concentration against *Enterobacteriaceae*, and total abundance of *Bifidobacteriaceae*, *Bacteroidaceae*, and *Clostridiales* in the gut lumen. A random predictor was included as a negative control. Predictor importance values were normalized within each time point between 0 and 1. The most influential variable in any column is black (value = 1.00) and progressively lighter shades indicate lower relative importance. Numeric values are overlaid for clarity. Starting from month 6 (DOL 180), endogenous immune responses against symbiotic commensals combined with ecological competition become the primary regulators of *Enterobacteriaceae* population, exerting stronger selection pressure than the endogenous SIgA (eSIgA) responses to *Enterobacteriaceae* itself, as seen from the decreasing importance of both the affinity and the concentration of eSIgA against *Enterobacteriaceae*. By DOL 720, importance values are relatively evenly distributed across predictors, consistent with the similarity in predictive power between total and SIgA-bound *Enterobacteriaceae* in Fig 3E. The data underlying this figure can be found in https://doi.org/10.5281/zenodo.15629746.
(TIF)

**S3 Fig. Effects of early life dysbiosis on microbiome composition and immune response dynamics.** Simulation of an early-life infection scenario by giving a selective advantage to the *Escherichia-Shigella* genus (compartment E) relative to Bifidobacteriaceae (compartment B) by reducing the human milk oligosaccharides (HMOs) to 25% of their normal concentration in breastmilk while keeping all other parameters constant (maternal SIgA levels and affinities, breastfeeding duration, and mixed feeding periods). Panel **A)** demonstrates the early life dysbiosis characterized by the overgrowth of E compared to data (magenta circle), which results in higher levels of sustained inflammation during early life (panel **B)**), leading to a more hyperreactive response against symbiotic commensals (magenta rectangle, panel **C)**). The data underlying this figure can be found in https://doi.org/10.5281/zenodo.15629746.
(TIF)

**S4 Fig. Illustration of model inputs without mixed feeding (MF) and differential impacts of EBF and MF durations in determining eSIgA affinity across a comprehensive range of feeding durations.** (**A**) Model inputs demonstrating a sharp transition from exclusive breastfeeding (EBF) to exclusive complementary feeding (ECF), with

no MF period in between, including normalized maternal secretory immunoglobulin A (mSIgA) concentration, human milk oligosaccharide (HMOs) and plant-derived polysaccharides (PDPs) calorie inputs, and timing of the endogenous immune system activation over time. EBF: exclusive breastfeeding; MF: mixed feeding; ECF: exclusive complementary feeding. (**B**) log (eSIgA Affinity) values at steady state for different combinations of EBF and MF durations, where the solid black line demonstrates the case of no MF followed by EBF. DOL: Day of life; E: *Enterobacteriaceae*; B: *Bifidobacteriaceae*; BC: *Bacteroidaceae*; C: *Clostridiales*. The data underlying this figure can be found in https://doi.org/10.5281/zenodo.15629746.
(TIF)

**S5 Fig. Comparison of various breastfeeding scenarios and their impact on endogenous affinity maturation.** Relative abundances in fecal samples, absolute abundances in the gut lumen, and temporal progression of average endogenous SIgA (eSIgA) affinities for (**A**)–(**C**) control, (**D**)–(**F**) hyperreactive mSIgA in breastmilk (BM), (**G**)–(**I**) SIgA deficient BM, (**J**)–(**L**) only exclusive complementary feeding (ECF), and (**M**)–(**O**) ECF with probiotic (*Bacteroidaceae* and *Clostridiales*) supplementation. DOL: Day of life; E: *Enterobacteriaceae*; B: *Bifidobacteriaceae*; BC: *Bacteroidaceae*; C: *Clostridiales*. The data underlying this figure can be found in https://doi.org/10.5281/zenodo.15629746.
(TIF)

**S6 Fig. Hypothetical scenario demonstrating the administration of TLR4 Antagonists to prevent inflammatory imprinting when mSIgA in breastmilk is insufficient.** Temporal progression of average endogenous SIgA (eSIgA) affinities for (**A**) control, (**B**) when mSIgA levels are 85% reduced, (**C**) when mSIgA levels are 85% reduced with TLR4 antagonists' administration (90% reduction in TLR4 stimulation). DOL: Day of life; E: *Enterobacteriaceae*; B: *Bifidobacteriaceae*; BC: *Bacteroidaceae*; C: *Clostridiales*. The data underlying this figure can be found in https://doi.org/10.5281/zenodo.15629746.
(TIF)

**S7 Fig. Visualization of the global sensitivity analysis.** Visualization of the global sensitivity analysis presented in S5 Table. Red dashed lines represent the 50th and 95th percentiles to distinguish the most influential parameters. The data underlying this figure can be found in https://doi.org/10.5281/zenodo.15629746.
(TIF)

**S8 Fig. Sensitivity analysis to the invasiveness parameter.** Endogenous versus maternal affinity levels (eSIgA vs. mSIgA) across taxonomic groups and varying levels of invasiveness. Each point represents the converged eSIgA affinity level for a given taxon under different levels of invasiveness, with shape denoting the degree of invasiveness (baseline, 2-fold, or 5-fold increase) and color indicating taxonomic group (*Bacteroidaceae*, *Bifidobacteriaceae*, *Clostridiales*). As invasiveness increases, uncoated bacteria more readily access GALT inductive sites, resulting in enhanced dendritic cell activation and a higher Tfh:Tfr ratio, which promotes increased eSIgA affinity. The data underlying this figure can be found in https://doi.org/10.5281/zenodo.15629746.
(TIF)

**S9 Fig. Sensitivity analysis to the ranges that determine the T cell help for different B cell fates.** Heatmaps of average endogenous SIgA (eSIgA) affinity values at the end of 735 days (2 years) for $0.05 \leq th_{range} \leq 0.75$ and $0.05 \leq th_{apop} \leq 0.75$ for **A)** *Enterobacteriaceae*, **B)** *Bifidobacteriaceae*, **C)** *Bacteroidaceae* and **D)** *Clostridiales*. Colors represent the magnitude of the eSIgA affinity values, with darker colors indicating larger values. Numerical values are indicated in each box. NA represents {$th_{range}$, $th_{apop}$} combinations with $1 - th_{range} \geq th_{apop}$ constraint. Baseline values ($th_{range} = 0.25$ and $th_{apop} = 0.25$) are indicated with the bold black boxes. Note that all affinity values targeting the symbiotic commensals (*Bifidobacteriaceae, Bacteroidaceae, Clostridiales*) are below 1, reflecting their predominantly masking behavior. This figure demonstrates the robustness of our affinity maturation model, showing that the exact numerical

values of $th_{apop}$ and $th_{range}$ do not affect the functional properties of the endogenous antibodies. The data underlying this figure can be found in https://doi.org/10.5281/zenodo.15629746.
(TIF)

**S10 Fig. Impact of early and delayed M cell opening on endogenous SIgA affinity maturation and *Enterobacteriaceae* load. A)** Average eSIgA Affinity values, **B)** Cumulative *Enterobacteriaceae* load in the gut lumen, and **C)** Cumulative *Enterobacteriaceae* load in the GALT inductive sites over the course of 2 years (735 days) for different delay durations of M cell opening relative to the baseline M cell opening time (DOL 129). DOL: Day of life. Early M cell opening increases *Enterobacteriaceae* antigen recovery in GALT inductive sites, potentially heightening susceptibility to enteric infections, yet triggers a more aggressive affinity maturation process against *Enterobacteriaceae*, reducing their cumulative burden in the gut lumen over time. Immune responses against symbiotic commensals remain largely unaffected. The data underlying this figure can be found in https://doi.org/10.5281/zenodo.15629746.
(TIF)

**S11 Fig. Impact of parameters $\tau^\delta$, $C_n$, and $c_n$ on endogenous SIgA (eSIgA) affinities.** Sensitivity of the average endogenous SIgA (eSIgA) affinity values at the end of 735 days (2 years) in response to **A)** the multiplier to adjust the incremental increase in the selection threshold during GC reactions ($\tau^\delta$), and **B)** the amplitude ($C_n$) and **C)** the decay rate ($c_n$) of the exponential function describing the diminishing pool of naïve T and B cells. Red dashed line marks the baseline values. Note that both axes are shown on logarithmic (base 10) scales. The data underlying this figure can be found in https://doi.org/10.5281/zenodo.15629746.
(TIF)

**S12 Fig. Comparison of predictive performance between cohort data and computational model. A)** Comparison of rankings for predictors between experimental data (blue) and model predictions (red). Connected points indicate the same predictor, with horizontal position showing rank order (1–5) in predictive importance. Identical ranking of paired points demonstrates the strong agreement between the model and the cohort data. **B)** Comparison of AUC (Area Under ROC Curve) values for cohort data (blue) and model predictions (red). Error bars represent the standard error. Calprotectin shows the highest predictive power in both datasets, while taxonomic groups demonstrate a moderate predictive performance. The dashed horizontal line at 0.5 represents the threshold for random prediction. The data underlying this figure can be found in https://doi.org/10.5281/zenodo.15629746.
(TIF)

**S1 Table. Taxonomic classification methods used across the studies for inference.** Summary of the taxonomic classification methods employed in studies used for inference, including details on the reference databases used, subjects, selection criteria, clustering methods, and bioinformatic tools.
(DOCX)

**S2 Table. Key taxonomic groups and their corresponding inoculation time, oxygen (O$_2$), and carbohydrate metabolism.**
(DOCX)

**S3 Table. Parameters with their corresponding descriptions, units, ranges, prior distributions, and model fit estimates.** A priori distributions for $\tau^c$ and $\tau^{new}$ are parametrized to reflect the significant contribution of somatic hypermutation (SHM) relative to the contribution of newly activated B cells in increasing BCR diversity. While some degree of affinity maturation can occur in the absence of germinal centers, the extent and efficiency of this process are substantially greater with SHM. SHM targets the variable regions of BCR genes for high-rate mutations, leading to a dramatic increase in BCR diversity and specificity. This mechanism far surpasses the initial diversity provided by newly activated B cells through

V(D)J recombination, crucially enhancing the immune system's ability to fine-tune and strengthen responses to specific antigens [6]. $\Gamma(\alpha,\beta)$ denotes the gamma distribution, where $\alpha$ and $\beta$ denote the shape and the rate parameter, respectively. $\beta(\alpha,\beta)$ denotes the beta distribution, where $\alpha$ and $\beta$ denote the shape parameters. $N(\mu,\sigma)$ denotes the normal distribution, where $\mu$ and $\sigma$ denote the mean and the standard deviation, respectively. $U(a,b)$ denotes the uniform distribution, where $a$ and $b$ denote the upper and lower bounds, respectively.
(DOCX)

**S4 Table. Additional assumptions implicit to the model structure.**
(DOCX)

**S5 Table. Results of the global sensitivity analysis.** Results of the global sensitivity analysis using the Morris method. This table lists the parameters included in the analysis, their respective descriptions, the ranges used for sampling, their inferred or calibrated values in the model, units, and the sensitivity metrics: $\mu$ (mean elementary effect, representing the overall influence of each parameter), $\mu^*$ (mean absolute elementary effect, indicating the magnitude of the parameter's impact irrespective of direction), and $\sigma$ (standard deviation of the elementary effects, representing the variability or nonlinearity in the parameter's effect on the output). The output variable considered in this analysis is the average endogenous SIgA affinity at DOL 735 against symbiotic commensals (*Bifidobacteriaceae*, *Bacteroidaceae*, and *Clostridiales*). The design includes 500 trajectories with 20 steps per trajectory, resulting in 10,000 total model evaluations.
(DOCX)

**S6 Table. Interpretation of the highest-ranked parameters in global sensitivity analysis and their impact on model dynamics.** This table presents parameters ranking above the 50th percentile (Q50) in our global sensitivity analysis (S7 Fig). For each parameter, we provide its biological interpretation, functional role within the model, and its quantitative influence on model outcomes.
(DOCX)

**S1 Text. Supplementary text.**
(DOCX)

## Acknowledgments

We are grateful to Prof. Dr. Emma Slack for their valuable time and insightful discussions, which significantly improved our work.

## Author contributions

**Conceptualization:** Burcu Tepekule, Ai Ing Lim, Charlotte Jessica E. Metcalf.

**Data curation:** Burcu Tepekule.

**Formal analysis:** Burcu Tepekule.

**Funding acquisition:** Burcu Tepekule, Charlotte Jessica E. Metcalf.

**Investigation:** Burcu Tepekule, Charlotte Jessica E. Metcalf.

**Methodology:** Burcu Tepekule, Charlotte Jessica E. Metcalf.

**Resources:** Charlotte Jessica E. Metcalf.

**Software:** Burcu Tepekule.

**Supervision:** Ai Ing Lim, Charlotte Jessica E. Metcalf.

**Visualization:** Burcu Tepekule.

**Writing – original draft:** Burcu Tepekule, Charlotte Jessica E. Metcalf.

**Writing – review & editing:** Burcu Tepekule, Ai Ing Lim, Charlotte Jessica E. Metcalf.

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
