## [Editor Report · Decision Letter 0]

Dear Burcu,

Thank you for submitting your revised manuscript entitled "The ontogeny of immune tolerance: a model of early-life secretory IgA - gut microbiome interactions" for consideration as a Research Article by PLOS Biology.

Your revision has now been evaluated by the PLOS Biology editorial staff as well as by the original academic editor with relevant expertise and I am writing to let you know that we would like to send your submission back to the previous reviewers.

However, before we can send your manuscript to the reviewers, we need you to complete your submission by providing the metadata that is required for full assessment. To this end, please login to Editorial Manager where you will find the paper in the 'Submissions Needing Revisions' folder on your homepage. Please click 'Revise Submission' from the Action Links and complete all additional questions in the submission questionnaire.

Once your full submission is complete, your paper will undergo a series of checks in preparation for peer review. After your manuscript has passed the checks it will be sent out for review. To provide the metadata for your submission, please Login to Editorial Manager (https://www.editorialmanager.com/pbiology) within two working days, i.e. by May 07 2025 11:59PM.

Kind regards,

Melissa

Melissa Vazquez Hernandez, Ph.D.

Associate Editor

PLOS Biology

---

## [Decision Letter · Decision Letter 1]

Dear Burcu,

Thank you for your patience while we considered your revised manuscript "The ontogeny of immune tolerance: a model of early-life secretory IgA - gut microbiome interactions" for publication as a Research Article at PLOS Biology after the previous Open Rejection. This revised version of your manuscript has been evaluated by the PLOS Biology editors, the Academic Editor and two of the original reviewers.

Based on the reviews, we are likely to accept this manuscript for publication, provided you satisfactorily address the remaining points raised by the reviewers. Please also make sure to address the following data and other policy-related requests.

a) We routinely suggest changes to titles to ensure maximum accessibility for a broad, non-specialist readership, and to ensure they reflect the contents of the paper. In this case, we would suggest a minor edit to the title, as follows. Please ensure you change both the manuscript file and the online submission system, as they need to match for final acceptance:

"A model of early-life interactions between the gut microbiome and adaptive immunity provides insights into the ontogeny of immune tolerance"

b) Could you please confirm that there is not grant number in some of the funding agencies?

c) Please note that per journal policy, the model system/species (human) studied should be clearly stated in the abstract of your manuscript.

d) During the discussion one of the reviewer suggested that you put Fig R2 from your response to R1 point 5 in the supplementary material, as they found the analysis useful. We would like to encourage this.

e) We do not have a word limit. Could you please move the section named Materials and Methods currently in the supplements in the main text? This would allow readers an easier access to the model. This also goes for the additional text and references place there.

Please supply the numerical values either in the a supplementary file or as a permanent DOI’d deposition for the following figures:

Figure R2ABC, 2A-I, 3A-E, 4AB, 5A-C, 6A-D, S1ABC, S3AB, S4A-O, S5ABC, S6, S7, S9, S10, S11, Table 1

g) Please cite the location of the data clearly in all relevant main and supplementary Figure legends, e.g. “The data underlying this Figure can be found in S1 Data” or “The data underlying this Figure can be found in https://doi.org/10.5281/zenodo.XXXXX”

h) Please ensure that your Data Statement in the submission system accurately describes where your data can be found and is in final format, as it will be published as written there.

i) Many thanks for providing the underlying code in GitHub. However, because Github depositions can be readily changed or deleted, please make a permanent DOI’d copy (e.g. in Zenodo) and provide this URL in the manuscript and Data Availability Statement.

We expect to receive your revised manuscript within two weeks.

*Published Peer Review History*

*Press*

Sincerely,

Melissa

Melissa Vazquez Hernandez, Ph.D.

Associate Editor

PLOS Biology

REVIEWERS' COMMENTS:

Reviewer #2 (Matthew Olm): Authors have adequately addressed all of my previous comments, and I am especially impressed with their efforts to make all imputed data public in the provided GitHub link. I believe the manuscript is now ready for publication.

Reviewer #3 (Mathias Hornef): The authors have made a major effort and responded to all my points and the points raised by the other reviewers seriously, extensively and in very much detail. They also included changes in the manuscript where appropriate. It remains unclear to me how big the effect of potential presently neglected or unknown influencing factors might be for the final outcome of the model. But I do believe that it may be worth trying and subsequent adaptations might ultimately allow and lead to a suitable approach. I therefore would opt to recommend publication.

The only way I could think of further enhancing interest and attention to a wider audience could be some sort of one or two example inputs with calculation to illustrate the potential value and applicability for wet lab researchers in this area.

---

## [Editor Report · Decision Letter 2]

Dear Burcu,

Thank you for the submission of your revised Research Article "A model of early-life interactions between the gut microbiome and adaptive immunity provides insights into the ontogeny of immune tolerance" for publication in PLOS Biology. On behalf of my colleagues and the Academic Editor, Yelizaveta Konnikova, I am pleased to say that we can in principle accept your manuscript for publication, provided you address any remaining formatting and reporting issues. These will be detailed in an email you should receive within 2-3 business days from our colleagues in the journal operations team; no action is required from you until then. Please note that we will not be able to formally accept your manuscript and schedule it for publication until you have completed any requested changes.

PRESS

Sincerely, 

Melissa

Melissa Vazquez Hernandez, Ph.D., Ph.D.

Associate Editor

PLOS Biology
